# Rivers in the sky, flooding on the ground: the role of atmospheric rivers for the inland flooding in central Europe

Monica Ionita[1], Viorica Nagavciuc[1,2] and Bin Guan[3,4]

[1] Alfred Wegener Institute for Polar and Marine Research, Bremerhaven, 27570, Germany
[2] Faculty of Forestry, Ştefan cel Mare University, Suceava, 720229, Romania
[3] Joint Institute for Regional Earth System Science and Engineering, University of California, Los Angeles, CA, USA,
[4] Jet Propulsion Laboratory, California Institute of Technology, Pasadena, CA, USA

*Correspondence to*: Monica Ionita (Monica.Ionita@awi.de)

**Abstract.** The role of large-scale atmospheric circulation and atmospheric rivers (ARs) in producing extreme flooding and heavy rainfall events in the lower part of Rhine River catchment area is examined in this study. Analysis of the largest 10 floods in the lower Rhine, between 1817–2015, show that all these extreme flood peaks have been preceded up to 7 days in advance by intense moisture transport from the tropical North Atlantic basin, in the form of narrow bands, also known as atmospheric rivers. Most of the ARs associated with these flood events are embedded in the trailing fronts of the extratropical cyclones. The typical large-scale atmospheric circulation leading to heavy rainfall and flooding in the lower Rhine is characterized by a low pressure center south of Greenland which migrates towards Europe and a stable high pressure center over the northern part of Africa and southern part of Europe and projects on the positive phase of the North Atlantic Oscillation. The days preceding the flood peaks, lower (upper) level convergence (divergence) is observed over the analyzed region, which indicates strong vertical motions and heavy rainfall. Vertically integrated water vapor transport (IVT) exceeds 600 kg·m$^{-1}$·s$^{-1}$ for the largest floods, marking these as very strong ARs. The results presented in this study offer new insights regarding the importance of moisture transport as driver of extreme flooding in the lower part of Rhine River catchment area and we show for the first time that ARs are an useful tool for the identification of potential damaging floods inland Europe.

## 1 Introduction

The intensity and frequency of precipitation extremes and floods have increased over the last decades in many parts of the world (Blöschl et al. 2015; Stadtherr et al. 2016). As a result, an increase in the flood hazard and its associated damages has become a major concern both for society and economy. In terms of economic losses, floods are the most widespread hazard at European level (Barredo 2009; Paprotny et al. 2018). Throughout the last decades Europe has been affected by numerous heavy rainfall events corroborated with damaging floods. Among the most costliest and damaging floods, at European level, we have: the 1993 and 1995 winter floods in France, Germany, Netherlands and Belgium (Chbab 1995; Fink et al. 1996; Engel 1997; Disse and Engel 2001); the 2000, 2007 and 2014 floods in U.K. (Kelman 2001; Posthumus et al. 2009; Muchan et al. 2015; Stevens et al. 2016); the 2002 and 2013 damaging floods in the Elbe river catchment area (Ulbrich et al. 2003a; Ulbrich et al. 2003b; Ionita et al. 2015a); the 2005 floods in the eastern part of Europe (Barredo 2007; Ionita 2015b) and the 2010 floods in central part of Europe (Bissolli et al. 2011), among others. These recent floods, recorded in different parts of the European continent, have shown that coping with floods is not trivial and for a better management and improvement of flood predictions it is necessary to improve our understanding of the underlying mechanisms of these extreme events. Taking into account the fact that climate change is expected to

lead to an intensification of the hydrological cycle and in particular the hydrological extremes (O'Gorman and Schneider 2009; Allan et al. 2014) it is imperative to properly understand the relationship between heavy rainfall events and floods and the prevailing large-scale atmospheric circulation on scales from planetary to mesoscale, in order to be able to provide skillful forecasts of upcoming floods in terms of time of occurrence, location and magnitude.

Flood risk management decisions and flood forecasting depends strongly on our understanding of the large scale drivers of hydrological variability (Lavers et al. 2014; DeFlorio et al. 2019; Guan and Waliser 2019). The timing, magnitude and duration of floods and heavy rainfall events depends on the hydroclimatic variability on different time scales ranging from hourly, daily, seasonal to interannual. This variability is connected to the large scale moisture transport on the entire atmospheric column, which in turn is controlled by different large scale teleconnection patterns such as the North Atlantic Oscillation (NAO), the Pacific North American Oscillation (PNA) and El Niño-Southern Oscillation (ENSO) (Guan et al. 2013; Paltan et al. 2017). Apart from these pre-defined large-scale teleconnection patterns, regional and local climates also modulate the water vapor transport, in the form of transient, narrow and elongated corridors, also known as atmospheric rivers (ARs) (Zhu and Newell 1994; Guan and Waliser 2019a; Ralph et al. 2018; Shields et al., 2018). ARs are responsible for ~90% of the poleward vertically integrated water vapor transport outside of the tropics and at any time there are at least 3 to 5 ARs around the globe (Zhu and Newell 1998; Guan and Waliser 2015). Their horizontal dimensions can be up to several thousands km long with an average width of ~ 500 km (Ralph et al. 2004b; Ralph and Dettinger 2011). The moisture associated with ARs can be visible on satellite images in the free troposphere as well as in the boundary level (Ralph et al. 2004a; Neiman et al. 2008), and their structure results from either local convergence along the cold front of extratropical cyclones (Dacre et al. 2015) or from transport from lower latitudes (Zhu and Newell 1998).

Different studies have linked the occurrence of ARs to extreme rainfall and flooding in different parts of the world, extending from arid/semi-arid areas to the polar regions (Ralph et al. 2019). For example, in California, ARs contribute to 30 – 50% of the river flow (Dettinger 2011) and they supply on average ~30% of the total precipitation in the west coast of U.S and Europe (Lavers and Villarini 2013a). On the other hand, heavy floods in California and Washington states have been linked to ARs occurrence (Ralph et al. 2006; Neiman et al. 2011), and similar results were later found in other west-coast areas. At European level ARs have been found to significantly influence heavy rainfall events over the Iberian Peninsula (Ramos et al. 2015; Eiras-Barca et al. 2016; Brands et al. 2017; Eiras-Barca et al. 2018), U.K. (Lavers et al. 2011; Lavers and Villarini 2015), Norway (Benedict et al. 2019; Hegdahl et al. 2020) and France (Lu et al. 2013). Lavers et al. (2011) have shown that U.K. extreme floods and precipitation are mainly driven by ARs, while Lu et al. (2013) related the extreme floods in the western part of France in January 1995 with tropical moisture exports in the form of ARs. Studies of ARs impacting the coastal European region found significant relationship between winter ARs and NAO (Lavers and Villarini 2013b). In the southern part of Europe ARs are concurrent with a positive NAO phase, whereas over the northern part of Europe, ARs are concurrent with a negative phase of NAO.

All the aforementioned studies, at European level, were conducted over coastal regions, where ARs make landfall, thus contributing substantially to extreme rainfall events and floods over these regions. Nevertheless, for the European mainland there are limited studies which show a direct link between ARs occurrence and heavy

rainfall events and flooding. For example, Paltan et al. (2017) have shown that 50% of the Rhine river floods can be related to ARs, but their study took into account all floods peaks which are exceeded 10% of the time, over the period 1979 - 2010. In this study, we want to explore the relationship between the 10 highest flood peaks (in terms of their magnitude) in the lower part of Rhine river catchment area and intense water vapor transport and the large-scale atmospheric circulation over the last 180 years, including the lead/lag relationship between the timing of flood peaks and the occurrence of AR conditions which has not been the focus of previous studies on AR-related flooding.

The objectives of this study are to (i) analyze from a hydrological point of view (e.g. daily hydrographs and flood magnitude) three of the most damaging winter floods (1925/26, 1993 and 1995) in the lower part of Rhine River catchment area; (ii) analyze the large-scale circulation preceding these extreme flood events; (iii) explore if/how the occurrence of ARs explains extreme flood peaks and (iv) use a cohesive long-term data sets (e.g. reanalysis products) to analyze the common drivers of 10 of the highest flood peaks (in terms of magnitude) recorded at Köln gauging station, situated in the lower Rhine.

The outline of the study is as follows. The basic features of the Rhine River catchment area are described in Section 2, while the data and methods are described in Section 3. The hydrometeorological situation in relationship with the most damaging floods is given in Section 4. The discussion and the main conclusions of the paper are presented in Section 5.

## 2 Study region and data

### 2.1 Catchment area

Rhine River ranks the 9[th] among the Eurasian rivers, having a total length of ~1250 km, a drainage area of ~185260 km$^2$ and an average streamflow of 2300 m$^3$/s. The catchment area of Rhine River covers 9 countries (Germany, Austria, Switzerland, Belgium, Luxembourg, Lichtenstein, Italy, France and the Netherlands) (Figure 1), while the river itself provides services for inland waterway transportation, drinking water, power generation, agriculture, tourism and urban sanitation (Uehlinger et al. 2009). The Rhine is Europe's most important inland waterway, transporting almost 200 million tons per year (approximately two-thirds of the European inland waterway volume) (Meißner et al. 2017).

Rhine River catchment area comprises different sub-basins (e.g. the Alpine Rhine, the High Rhine, the Upper Rhine, the Middle Rhine and the Lower Rhine), influenced by different meteorological conditions. As such, the Rhine river has a complex runoff regime, with a summer maximum on the Alpine, High and Upper Rhine, and a winter maximum on the Middle and Lower Rhine, influenced by tributaries Main and Moselle (Belz 2010; Pfeiffer and Ionita 2017). Extreme floods events in the Rhine region can be divided in two classes according to their hydrometeorological causes (Belz et al. 2007): i) winter/spring floods, triggered by warm air intrusions and snow melt, occurring mostly in the flatlands (e.g. Lower Rhine) and ii) summer floods, triggered by heavy rainfall and low snow melt in the Alpine region. Documentary and historical records, since 1000AD, do not list a single flood event which occurred simultaneously in all sub-basins of the Rhine river (Disse and Engle, 2001).

The major tributaries of Rhine river, on the German side are: Neckar, Main and Moselle. All the tributaries are characterized by a pluvial regime, with the mean runoff reaching the highest values in the winter months and the minimum in August- September. Throughout the 20[th] century, flood peaks show an upward trend in the Alpine and pre-Alpine Rhine basin (Belz et al. 2007). In the Middle and Lower Rhine, extreme floods due to enhanced

rainfall sometimes combined with snowmelt, have increased in intensity, especially during the winter months, in the second part of the 20[th] century (Belz et al. 2007). The increasing trend in the flood peaks, in the Middle and Lower Rhine basin, is largely due to the contribution from the Moselle River where flood waves with peaks up to 4200 m$^3$/s can occur in winter months (annual mean average discharge at Cochem gauging station = 313  m$^3$/s). Min (2006) has shown that the basin-averaged precipitation for the Moselle basin shows a significant increase since 1980, accompanied by a tendency towards more frequent intense precipitation events (e.g. exceeding 10mm/day) in the winter half-year (November–April). The overall increase in winter precipitation in the catchments of the lower Rhine tributaries, has caused an increase of ~10% of the mean discharge at Lobith gauging station during the 20[th] century.

Flooding along the Rhine River basin is a natural phenomenon. Nevertheless, flood risk, especially in the Upper Rhine has been highly increased by river training. The Upper Rhine was subjected to heavy river training from the beginning of the 19[th] century until 1977. As a result of this river training, the flood risk downstream has considerably increased due to a shortening of the river course, a reduction of potential floodplains by constructing dikes directly on the summer river bed, increased velocity of waves and the overlapping with flood waves from the tributaries. For Köln gauging station, Pinter et al. (2006) have shown that river engineering had an insignificant effect on the flood magnification throughout the 20[th] century. The main driver of the flood magnification at Köln gauging station has been found to be an increase in the precipitation over the Rhine basin. A small contribution to the flood magnification in the lower Rhine comes also from land use and industrialization of the German agriculture.

## 2.2 Data

The main variable analyzed in this study is the daily streamflow data at Köln gauging station situated in the lower part of the Rhine catchment area (Figure 1). Köln gauging station is one of the  most important gauges along the Rhine; at this point starts one of the most populated area over Rhine basin - North Rhine-Westphalia, respectively. In addition, before Köln gauging station there is a confluence point for three of the main tributaries of the lower Rhine: Lahn, Moselle and Sieg. We choose this gauging station for the current analysis due to the availability of daily streamflow data (1817 – 2019), the importance of the Lower Rhine both from an economical and societal point of view and because the river training effects are not so significant in this part of Rhine's catchment area. The daily streamflow data was provided by the German Hydrological Institute (www.bfg.de).

To identify the flood events, we extracted, from the daily streamflow time series at Köln gauging station, the top ten daily streamflow values over the period 1817 – 2019 . We have compared our flood events with the ones from the information platform of the German Hydrological Institute (http://undine.bafg.de/rhein/rheingebiet.html). We have restricted our analysis just over the winter months (November – March), due to the  fact that summer floods tend to be produced by different mechanisms (e.g. convective precipitation). In the lower Rhine, more than 80% of the flooding events occur during the period November – March (Figure S1). For the top 10 winter flood events over the lower Rhine River basin (Table 1), we have extracted also daily precipitation and daily mean air temperature at Trier weather station, which is situated on the main course of Moselle River, one of the most important tributaries of Rhine River. We choose Trier station due to its length (1907 – 2019) and due to the fact that most of the floods on the lower basin of Rhine river are mainly influenced by the input from the Moselle

river (Figure 1). The daily precipitation and daily mean air temperature data were extracted from the ftp server of

the German Weather Service (ftp://opendata.dwd.de/climate_environment/CDC/).

For the large-scale atmospheric circulation, we have used the daily sea level pressure (SLP), zonal and meridional

wind at 500mb level, geopotential height at 500mb level, potential vorticity, air temperature at 850mb level,

specific humidity and surface pressure from the 20th Century Reanalysis data V3 (Slivinski et al. 2019). The 20th

Century Reanalysis data V3 uses a state-of-the-art data assimilation system and surface pressure observations, it

has 64 vertical levels and 80 ensemble members. The output of this reanalysis product is a 4D global atmospheric

dataset spanning the period 1836 to 2015. The resolution of the data set is ~1° x 1° (Slivinski et al. 2019).

The vertically integrated water vapor transport (IVT) (Peixoto and Oort 1992) is calculated through zonal wind

(u), meridional wind (v) and specific humidity (q), from the 20th Century Reanalysis v3 data set. IVT vectors for

latitude ($\phi$) and longitude ($\lambda$) are defined as follows:

$$\vec{Q}(\lambda,\phi,t) = Q_\lambda \vec{i} + Q_\phi \vec{j}$$

Where zonal ($Q_\lambda$) and meridional ($Q_\Phi$) components of Q are given by:

$$Q_\lambda = - \int_{1000}^{300} qu \frac{dp}{g}$$

$$Q_\phi = - \int_{1000}^{300} qv \frac{dp}{g}$$


where $u$ is the zonal wind component, $v$ is the meridional wind component, $q$ is the specific humidity and $g$ is the

gravitational constant. The ARs are identified using the vertically integrated water vapor transport (IVT) between

the 1000 and 300 hPa levels. The methodology used in this study to define an AR is based on the global detection

algorithm of Guan and Waliser (2015). We consider an AR if the IVT exceeds an intensity threshold (e.g. the

local 85th percentile), if it has a minimum value of 100 kg·m$^{-1}$·s$^{-1}$ and a length of at least 2000 km, among other

considerations detailed in Guan and Waliser (2015). Performance of this AR detection algorithm has been

validated against dropsonde observations over the north-eastern Pacific (Guan et al. 2017), and results based on

this global algorithm were found to agree well with algorithms independently developed for three specific regions

(Guan and Waliser 2015).

Using the 20th Century Reanalysis v3 data set, we computed also the integrated water vapor (IWV) which is

defined as follows:

$$IWV = \int_{1000}^{300} q \frac{dp}{g}$$

To analyze the spatial distribution of the daily precipitation amount during the days prior the flood peaks we have

used two gridded datasets: daily precipitation data from the EOBS-v20e data set (Cornes et al. 2018), with a

spatial resolution of 0.25° x 0.25° and the REGNIE daily precipitation dataset (Rauthe et al. 2013) with a spatial

resolution of 1km x 1km. The E-OBS data set cover the whole European region, while the REGNIE dataset is

restricted to Germany.


## 3 Results

### 3.1 Hydrometeorological situation - 1925/26

The last week of November 1925 was characterized by heavy snowfall in the western part of Germany and the lower areas of the Rhine River basin (Soldan 1927). After a short period of dry and cold days, at the beginning of the second week of December it began to thaw, until mid-December. In parts of the Rhine area, heavy snowfall corroborated with extremely low temperatures were recorded over the period $12 - 18^{th}$ of December 1925 (Figure S2 and Figure S3). After this period of heavy snowfall and low temperatures, warm and humid air masses penetrated from the south-west on the $26^{th}$ of December. The warm and humid air slid onto the cold air between the mountain areas of the Rhine region and caused extremely high rainfall in large parts of the Rhine catchment area between $27^{th}$ until $29^{th}$ of December 1925 (Figure S4). From the $27^{th}$ until $31^{st}$ of December heavy rainfall affected large parts of the Moselle's catchment area and the lower part of Rhine River basin. The highest rainfall amount, at Trier station, was recorded on the $27^{th}$ of December (27mm) (Figure 2a). The cumulated rainfall amount over the period $27 - 31$ December was 92.5mm. Overall, in December 1925 the total rainfall amount over the lower part of Rhine Rivers basin was on average almost double compared to the climatological mean (Figure S5a). These extreme rainfall events in the last week of December 1925, were driven by a dipole-like structure in the SLP filed with a deep low pressure system (960 hPa) centered south of Greenland and a high pressure system (1025 hPa) centered over the northern part of Africa (Figure 3a). This dipole-like structure started to develop on the $25^{th}$ of December and persisted until the $31^{st}$ of December, slightly shifting its centers (Figure 3). This circulation structure led to a pronounced south-westerly air flow over France and western part of Germany, associated with a narrow band of atmospheric moisture extending from the U.S coast up to central part of Europe. The narrow band of moisture was particular active on the $27^{th}$, $28^{th}$, $29^{th}$ and $30^{th}$ of December (Figure 3c, 3d, 3e and 3f), leading to heavy rainfall over northern part of France and western part of Germany (Figure S4). The ascend of the band of moisture in the warm sector of the extra-tropical cyclone and the rising over the mountain areas in the Moselle catchment area (southern Vosges Mountains in France, and Hunsrück and Eifel Mountains in Germany), led to heavy rainfall and thus to high flood peaks some days later in the lower part of Rhine River basin. At Trier gauging station, the flood peak reached its third highest value over the observational record (1817 – 2020, 3600 m$^3$/s) on the $31^{st}$ of December 1925. During the days with extreme rainfall and flood peaks at Trier station (27 – 30.12.1925, Figure S4) the maximum of the water vapor flux within the plume exceeded 700 kg·s$^{-1}$·m$^{-1}$ (Figure 3, Table 2), while at day -1 and day 0 of the flood peak at Köln gauging station the vapor flux weakened to ~400 kg·s$^{-1}$·m$^{-1}$. The arrival of the ARs towards the western part of Europe coincides with a sharp increase in the temperature over the western part of Europe (Figure S6), strong advection of moisture and heavy rainfall towards our analyzed region.

Looking more into detail into the dynamic fields of IWV during the days with heavy rainfall events, one can identify a distant source of moisture. During the days of rainfall events, recorded at Trier station, a long and narrow band of IWV is transported from the sub-tropical latitudes, passing the North Atlantic Ocean and reaching the western part of Europe (Figure 4). This narrow band of moisture is transported by a southwestern wind, at 900 hPa level, above 20m/s (Figure S7 – left column). This combination of wind and IWV, concentrated in such narrow bands suggests the presence of an AR, in agreement with the results obtained by using the AR tracking algorithm of Guan and Waliser (2015) (Figure 3).

The synoptic evolution of the tropopause level flow is analyzed using the dynamical tropopause on the 330K isentropic level (Figure 4). Previous studies (Browning et al. 1997; Froidevaux and Martius 2016) have shown that flooding and extreme rainfall are linked with upper-level troughs associated with the presence of elongated intrusions of air, the so-called potential vorticity (PV) streamers. Southern intrusion of air with high PV in the lower stratosphere or higher troposphere are corroborated with lowering of the dynamical tropopause, intense vertical motions, cyclogenesis and heavy rainfall (Krichak et al. 2014; Rimbu et al. 2020). The days prior to the flood peak, featuring extreme rainfall at Trier gauging station, are associated with high PV values (PV >2PVU) over the analyzed region (Figure 4). The axis of the IWV field follows the nonlinear behavior of the PV and the maximum of the IWV is situated in the overturning of the 2PVU contour line (Figure 4). The shape of the 2PVU contour lines indicates the presence of an anticyclonic Rossby wave breaking (Payne and Magnusdottir 2014). AR activity is also linked with the breaking of the midlatitude Rossby waves (RWB), which can be either cyclonic or anticyclonic (Payne and Magnusdottir 2014). In this respect, we have found that all the ARs that have passed the western part of Europe, prior to the flood peak were associated with an anticyclonic Rossby wave breaking (ARWB) (Table 2), which is in agreement with the study of Zavadoff and Kirtman (2020) who have shown that ARs over the western part of Europe are linked mostly with ARWB, and in broad consistency with (Hu et al. 2017) who found most of the AR landfalls along the northern coast of western US were associated with ARWB.

In order to further analyze the dynamical drivers of extreme flood events we computed the divergence field for the upper (300 hPa) and lower (900 hPa) levels wind speed. The wind speed and its associated divergence/convergence at the 900 (300) hPa level are shown in Figure S7. The upper and lower level analysis (Figure S7 – right column) indicates the presence of an upper level jet branch, shifted southwards and stretching from the central North Atlantic basin until the central part of Europe and an area with upper level divergence over our analyzed region (contour lines in Figure S7), which is an indicator of deep convection (Hoskins et al. 1978; Krichak et al. 2014). In addition, at lower levels, we can observe an intense low-level jet is present with a similar orientation as the upper level jet (Figure S7 – left column) over the analyzed region. The upper level divergence and lower level convergence over our analyzed region, are an indication of strong vertical motions and heavy rainfall (Hoskins et al. 1978). The days characterized by rainfall episodes (e.g. 27 – 31.12.1925 – Figure 2a) are all associated with a southward shift of the polar front, convergence over the catchment area (dashed lines in Figure S7) and enhanced moisture transport in narrow band stretching from the tropical Atlantic until our analyzed region (Figure 4).

From a hydrological perspective, the thaw that started in the second week of December 1925 brought the first flood peaks in the Rhine River and its tributaries (especially the Neckar and Mosel), which peaked in the middle of the month (Figure 2b). The subsequent frost period reduced the water flow, before the renewed thaw caused rising peaks again in the third week of December. Over this period of time, most of the Rhine River tributaries (e.g. the Aare, Murg, Kinzig, Neckar, Lahn) and especially the Moselle brought relatively large volumes of water to the Rhine (Soldan 1927). This caused the flood peaks in the Middle and Lower Rhine to rise abruptly. The subsequent brief cold snap caused the water levels to drop again from 23[rd] of December, before the abrupt weather change on 26[th] of December, which led to the outstanding flood event. The rapid ascent of the Rhine began on the 27[th] of December. On the Upper Rhine, the flood peak of the Ill river merged with that of the Rhine

on the 30th of December (Soldan 1927). The water of the Moselle reached the Rhine on early January 1. Dike breaks occurred above Köln and at Neuss. In the Prussian Rhine province, more than 28000 houses and 2500 businesses were flooded, and more than 13500 apartments had to be cleared. The most severely damaged cities were Köln and Koblenz with 72000 and 14000 people affected, respectively. Agricultural damage was also significant as 74000 hectares of land were under water. Arable crops and crop stocks were destroyed and gravel and sand were deposited on the cultivated areas. The damage to hydraulic engineering systems on the Rhine, Mosel and Ruhr was put at ~284000 €. The damage to the traffic facilities outside the rivers was rather small given the size of the flood. The total damage was estimated to ~75 million € for the Prussian Rhine region (Soldan 1927).

**3.2 Hydrometeorological situation - 1993**

November 1993 and the first week of December 1993 were characterized by relatively reduced amounts of rainfall over Rhine River catchment area (Bornefeld 1994). Between 8th to 24th of December, the general weather situation was characterized by several Atlantic low pressure systems (heavy storm with hurricane gusts on December 9) and a stable high pressure system in front of the North African continent, which led to frequent rainfall, sometimes heavy rain over the northern part of France and western part of Germany (Deutscher Wetterdienst 1994). The maximum amount of precipitation, at Trier station, was recorded on the 20th of December 1993 (33.3mm) (Figure 5a). Overall, in the period from 8th to 20th of December 1993, in the catchment areas of Neckar, Nahe and Mosel, as well as in large parts of the central Upper Rhine and central Middle Rhine, more than double of the long-term December precipitation, was recorded (Figure S5a and S8). The extratropical cyclones in the North Atlantic persisted several days pushing a narrow band of moisture towards central Europe (Figure 6). High rates of IVT (up to 800 kg m$^{-1}$ s$^{-1}$) were recorded from the 18th until 21st of December (Figure 6b, 6c and 6d). Over lower Rhine, the maximum of the IVT reached values up to ~620 kg·s$^{-1}$·m$^{-1}$ (Table 2). At the same time, warm, humid air entered the Rhine area, which led to an extraordinary increase in temperature (Figure 5a). After a rainless 18th of December, the 19th and 20th of December were characterized by heavy rainfall events over most of Rhine's River catchment area (Figure 5a). The days characterized by heavy rainfall over our analyzed region (Figure S8) are associated with a narrow band of IWV (Figure 4), stretching from the sub-tropical North Atlantic basin until the central part of Europe, strong winds directed towards the western part of Europe and lower (upper) level converge (divergence) (Figure S10). This large-scale pattern is favorable of strong vertical motions and heavy rainfall over our analyzed region. As in the case of the 1925/26 flood peak, the axis of the IWV follows the path of the 2PVU contour line (Figure 7) and exceeds 20mm for all the days with heavy rainfall at Trier station (Figure 5). For the 1993 flood peaks, the days prior to the flood were associated with anticyclonic RWB (Table 2) which led to sharp meridional gradients of PV over Europe (Figure 7). Positive upper level PV anomalies affect the structure of the atmosphere below them (Schlemmer et al. 2010), such that cold air and reduced static stability are found below the positive PV anomalies (Figure S9). The PV streamer observed over the period 18 – 21.12.1993 (black contour line Figure 7) was associated with warm air advection towards the western part of Europe (Figure S9) and cold air advection over the eastern part of Europe, which led to snow melt and heavy rainfall over the lower Rhine catchment area (Figure 5 and S9).

The precipitation in the first half of December 1993 caused the soil to become saturated with water, so that the subsequent rain had an immediate drainage effect (Soldan 1927). However, a noteworthy flood wave only developed in the Rhine from the inflow of the Neckar, which led to a significant flood and reached its peak water level on December 22 at the Rockenau station (Engel et al. 1994). The highest flow at Kaub gauging station dates to December 23[rd]. An extraordinarily flood peak developed in the Moselle, and the highest daily streamflow of the century was recorded at the Cochem gauging station on the 22[nd] of December (4020 $m^3$/s). This flood peak in the Moselle merged with the Rhine on December 23, leading to one of the largest known flood waves of the Rhine downstream of Koblenz (Figure 5b). The combined flood peak passed through Andernach on 23[rd] of December 1993 and reached Köln gauging station on 24[th] of December (10600 $m^3$/s) (Figure 5b), where the flood protection wall of the old town was flooded for about 70 hours, and Rees and Emmerich cities on the following day.

The Christmas flood in the Rhine area caused several human losses and required evacuations in many cities. In Baden-Württemberg (Neckar area), Rhineland-Palatinate and North Rhine-Westphalia high building damage occurred. In Koblenz, almost a quarter of the built-up area of the city was flooded, while 10000 inhabitants and ~ 4000 houses were directly affected by the flood. In North Rhine-Westphalia, the number of damaged households was considerable, especially in Königswinter, Bonn and especially in Köln. In Köln, over 4500 households had a direct flood damage and another 9000 households suffered damage from increased groundwater. The damage to the federal shipping routes was estimated to be ~9,8 million €. Due to the flooding, the shipping had to be completely stopped on Neckar and Saar for 9 days, on the Mosel 12 days and on the Rhine from Koblenz to the Dutch border for 7 days. The lost transport revenues were estimated at over 38 million €. A total of approximately 1,4 billion € was damaged in the whole Rhine regions affected by this flood event (Münchener Rückversicherungs-Gesellschaft 1999).

### 3.3 Hydrometeorological situation - 1995

After local heavy rain in the higher low mountain regions at the end of December 1994, several periods of precipitation in January 1995 brought heavy rainfall to the entire Rhine region (Engel 1999). The cold spell in the first week of January 1995 (Figure 8a) led to snow fall in the middle part of Rhine River. In the Alpine region there were almost 100 cm of snow recorded at the end of the first week of January 1995. In the second week of January 1995, a north-westerly flow led to daily showers on top of the accumulated snow (Engel 1999). Thus, the rainfall corroborated with the snow melt, due the temperature increase, led to small flood peaks along the Rhine and its tributaries (Figure 8b). The situation became exceptional starting with the 22[nd] of January 1995. The rainfall episodes began on the evening of the 21[st] of January and last ~30 hours. The total precipitation recorded in January 1995 (168 mm), overaged over the German part of the Rhine basin, is the highest one recorded over the period 1881 – 2019 for the month of January (Figure S5b). The extreme rainfall episode was triggered by a frontal system which deepened into a low pressure system over the western part of Europe. On the 22[nd] of January 1995, Trier meteorological station recorded the highest daily precipitation amount (49.6mm) over the last 70 years for the month of January. This event led to an increase in the water levels of Moselle river to a flood peak of 2880 $m^3$/s at Trier station and 3410 $m^3$/s at Cochem gauging station, on the 23[rd] of January 1995 (Figure 8b). On the 25[th] of January 1995, another exceptional rainfall event was recorded, with values up to 55mm in 24

hours over large area in the Moselle and Rhine catchment areas (Figure S11). This event was triggered by an AR event, with a magnitude of ~600 kg·s$^{-1}$·m$^{-1}$ (Figure 9b) and a westerly flow of the large-scale atmospheric circulation characterized by a deep low pressure system over Scandinavia and a high pressure system over northern part of Africa. Between 26[th] to 29[th] of January 1995, small frontal waves driven by the low pressure system south of Greenland (Figure 9c-f) led to more rainfall episodes (Figure 8a) over Moselle's catchment area and large parts of Rhine River catchment area (Figure S11). These frontal systems were corroborated with ARs stretching from the sub-tropical North Atlantic basin and bringing moisture to the western part of Europe (Figure 9e and 9f). The prevailing large-scale atmospheric circulation, during the days characterized by enhanced rainfall over large area of Rhine's catchment area (Figure S11), featured narrow bands of moisture transport from the sub-tropical North Atlantic basin towards the western part of Europe (Figure 10) and enhanced lower level convergence over central part of Europe (Figure S13). The upper level large-scale atmospheric circulation was characterized by divergence over Rhine's catchment area, ascending motions and heavy rainfall (Figure S13). As in the case of the 1925/26 and 1993 flood peaks, the axis of the IWV follows the path of the 2PVU contour line (Figure 10) and exceeds 20mm for all the days with heavy rainfall at Trier station (Figure 10). For the 1995 flood peaks, the days prior to the flood were associated with anticyclonic RWB (Table 3) which led to sharp meridional gradients of PV over Europe (Figure 10) and the advection of warm air advection towards the western part of Europe (Figure S12) and cold air over the eastern part of Europe. The combination of warm aid advection and intense ARs, lead to snow melt and heavy rainfall over the Rhine catchment area and central part of Germany (Figure S11).

As a result of the repeated rainfall episodes, the catchment area in the Middle and Lower Rhine as well as the tributaries Main, Nahe, Mosel and Sieg saw a steep increase in daily streamflow (Engel 1999). The water inflow from the Main and Nahe rivers resulted in a significant increase in the flood wave at the Mainz and Kaub gauges, where the Christmas flooding from 1993 was exceeded. On 30[th] of January 1995, after the flooding of the Moselle began, the flood peaks of the Rhine reached 10700 m³/s at Köln gauging station (the second highest daily streamflow recorded over a period of 200 years). Due to the flood inflow of the Sieg, the flood peak at Köln gauging station was further increased, reaching with a water level 6 cm higher than during the Christmas flood in 1993 (Engel 1999). At the Rees gauging station the peak of the flood wave was 11300 m³/s and was above the Christmas flood in 1993 (10600 m³/s) and only just below the highest known flood peak in January 1926 (11.700 m³/s). Overall, the extreme floods of January 1995 were mainly triggered by long lasting rainfall episodes driven by frontal systems from the North Atlantic basin, a high frequency of AR events and intense moisture transport from the sub-tropical North Atlantic basin until the western part of Europe, as well as a southward shift of the polar front and upper (lower) level divergence (convergence) over the analyzed region.

The 1995 floods had huge consequences both for society and economy in Germany and the Netherlands. In numerous cities and towns on the Rhine and the tributaries, streets and houses were flooded, power outages and damage to the infrastructure occurred and 5 people were killed (Münchener Rückversicherungs-Gesellschaft 1999). The total monetary damage in the German Rhine catchment area estimated to be ~398 million €. Due to the exceeding of the highest navigable water levels, shipping had to be temporarily suspended on individual sections of the and the monetary losses for the shipping related companies was ~ 36 million €. For the Köln city alone, the damage was around 47 million € (half as much as in the 1993 Christmas flood) and 4000 people were

directly affected by the floods. In the Netherlands, at least 4 people were killed and ~250000 people had to be evacuated because of dike breaches and extensive flooding of polders and large parts of cities were submerged between 30 January and 1 February 1995 – from the Limburg region south of Nijmegen and from Zeeland, around Rotterdam, Europe's largest port (Münchener Rückversicherungs-Gesellschaft 1999).

**3.4 Composite events**

The crucial role that ARs have in preceding extreme flooding in the lower part of Rhine river catchment area is highlighted also by analyzing the IVT, SLP and the AR origin with different time lags (0 – 7 days) for the 10 highest flood peaks measured at Köln gauging station. The occurrence date and the magnitude of each flood peak are shown in Table 1. The composite of all flood peaks shows that all of them are preceded up to 7 days by a plume of moisture, in the shape of an AR, accompanied by a deep low pressure center over the British Isles and a high pressure center over the Iberian Peninsula (Figure 11). The maximum of the IVT, averaged over the 10 floods peaks, reaches values of up to 500 kg·s$^{-1}$·m$^{-1}$. The ARs associated with heavy winter floods in the lower part of the Rhine River basin have an elongated shape, thus confirming the AR geometrical criterion of being at least 2000 km long and less than 1000 km wide (Ralph et al. 2004; Neiman et al. 2008).

Snapshots of the evolution of the AR axis, indicating the development and propagation of the AR prior to the occurrence date of the top 10 flood peaks, over the last 180 years, are shown in Figure 12. The evolution of ARs for the different snapshots demonstrate further the importance of moisture transport from the North Atlantic Ocean in producing damaging floods in the lower part of Rhine river basin. In all cases the moisture transport is directed towards the north-western part of France penetrating until the western part of Germany. The axis of the ARs is strongly influenced by the dipole-like structure in the SLP field, with a migrating deep low over the central part of the North Atlantic and a high pressure system over the northern part of Africa and Iberian Peninsula. For all the 10 analyzed cases there was at least one AR preceding the flood peak with a lag varying between 2 to 7 days. The maximum of the IVT, over the catchment area of Rhine, is reached 4 up to 6 days prior each flood peak (Table 2).

In a long term perspective, there is a significant positive trend in the AR occurrence rate (~0.9 days /decade) over the analyzed region (Figure 13a) corroborated with a positive, but not significant, trend in the precipitation averaged over the catchment area of Rhine River over the winter months (November – March) (Figure 13b). The highest magnitude of ARs reaching up to the lower Rhine was recorded from 1990s onwards. The increase in the number of ARs reaching up to the lower Rhine and the positive trend in precipitation does not necessary mean an increase in the annual maxima of the daily streamflow at Köln gauging station over the period 1836 – 2015 (Figure 13c). The lack of change in the amplitude of the annual maxima at Köln gauging station might be influenced also by the lack of snow cover, which is a pre-requisite for extreme flood peaks, like in the case of the years 1925/26, 1993 and 1995. At country level, snow days have decreased uniformly at a rate of 0.5 days/year in the recent past (Kreyling 2011), this trend being projected to continue to a point where significant parts of Germany will no longer regularly experience snow cover.

### 3.5 NAO and Circulation types

Over the analyzed region, the floods events of 1970, 1988, 1993 and 1995 were all preceded by days with a positive NAO index (not shown). In the case of the 1993 flood event, the daily NAO index was characterized by positive values for almost two consecutive months (26.10.1993 – 22.12.1993). For the whole analyzed period (1836 -2019), just monthly data are available for the NAO index. A visual inspection of the NAO index during the month of each flood event, indicates that in 9 out of the 10 extreme floods events, NAO was in a positive phase. The only exception is the flood peak from 31.03.1845 (March NAO = -0.54). The values of the NAO index for each flood peak are shown in Table 3. Overall, extreme flood peaks, in the lower Rhine region seem to be preceded by a positive phase of the NAO. A positive phase of NAO is, in general, associated with an increased chance of higher rainfall in northwest Europe and lower rainfall in southern Europe. The long lasting positive NAO prior to most of the flood peaks, might be one of the main driver behind the high magnitude of the ARs. Zavadoff and Kirtman (2020) have shown that long lasting phases of NAO have a more significant influence of the ARs distribution when compared with short-lived NAO phases.

Looking more into detail at country level, the influence of NAO on the hydroclimate of Germany was found to be very complex (Riaz et al. 2017). The relationship between winter NAO and precipitation was found to be rather week in the southern part of Germany, but statistically significant in the northern part. Riaz et al. (2017) have shown that the precipitation over Germany is mainly influenced by the position of the Icelandic Low, independent of the strength and position of the Azores High. Thus by using the state of the art definition of the NAO, namely the difference in the sea level pressure between Iceland an Azores, one cannot capture the real influence of NAO on the central European climate. A way to tackle this issue is to look at synoptic scale circulation types (e.g. Großwetterlage, (Hess and Brezowsky 1952)). When looking at particular circulation types (e.g. Großwetterlage) the extreme flood peaks from 1881 onwards are all preceded mainly by days featuring either Cyclonic west wind (WZ) or Southern west wind (WS) circulation types. Both types represent zonal circulation types (westerly flow), associated with extreme precipitation and floods over Germany (Petrow et al. 2009). For example, the 1993 and 1995 flood events were preceded by days featuring just the WZ circulation type (Table 3). Caspary (1995) has shown that over the period 1926 – 1996, nearly all flood events in the Upper Danube River basin have been caused by WZ circulation type. In their study, Petrow et al. (2009) have found that 62% of the maximum discharges in the basins situated in the western part of Germany are triggered by the circulation patterns: WZ, WS, SWZ and NWZ. In this study we have found that the WZ circulation type was present, in the days prior to the flood peaks, in 7 out of 8 of analyzed extreme flood events over the period 1881 – 2019 (Table 2). For the 1993 and 1995 flood events, only the WZ circulation type was present the days prior to the flood peak. Overall, the occurrence of these CTs (WZ and WS) seem to be a pre-requisite for extreme flood events in central part of Europe (Caspary 1995; Petrow et al. 2009).

### 4. Discussion and Conclusions

The variability of European precipitation, in winter, is strongly affected by enhanced moisture transport from the sub-tropical North Atlantic basin (Lu et al. 2013; Lavers and Villarini 2015). Overall, ARs are responsible for ~20 - 30% of all recorded precipitation in regions situated in the western part of Europe (mainly France and Iberian Peninsula) (Gimeno et al. 2016). While ARs are essential ingredients in producing heavy rainfall events

and flooding over the coastal areas of the European continent (e.g. Portugal, Spain, France, Norway) little is known about their influence on the precipitation and flood events inland Europe (Gimeno et al. 2016). Although there have been numerous studies linking ARs with floods and heavy precipitation, over large parts of the world (Neiman et al., 2008; Lavers et al. 2011; Dettinger 2011; Lavers and Villarini 2013b; Marengo et al. 2016; Paltan et al. 2017; Vázquez et al. 2017, Benedict et al. 2019; Guan and Waliser 2019b; among others) this is the first study in which ARs are linked with specific events of extreme flooding inland Europe, more specific over the lower part of Rhine catchment area, which is one of the biggest rivers in Europe. The lower part of Rhine catchment area is dominated by winter floods, which are often caused by westerly, southwesterly and north-westerly large-scale circulation types (Beurton and Thieken 2009).

In this study we have shown that extreme floods in winter, based on their magnitude, occur predominantly during mild and wet episodes associated with a southwards shift of the polar front and frontal systems moving from the North Atlantic basin towards Europe, corroborated with intense moisture transport from the sub-tropical North Atlantic basin until the western part of Europe. From a hydrological point of view (e.g. flood peak magnitude), the 1925/26 flood was the worst flood of the 20th century at Köln gauging station. However, the total volume of water was much higher for the 1993 flood than in 1925/1926 (Engel et al. 1999). The Rhine transported ~ 60% more water during the 1993 flood compared to the 1925/26 flood event. The total damage caused by the Rhine flood in 1993 was estimated at ~1.4 billion euros, however, the damage caused by the Christmas floods in 1995 was only about half as great as in 1993, due the prevention measures implemented by the Rhine River countries after the 1993 flood event and due to earlier flood warnings.

Although the mechanism behind each individual extreme flood can be rather different (e.g. heavy snowfall, a sharp increase in the mean air temperature followed by thawing and/or just extreme rainfall events), all the analyzed floods have one thing in common: the heavy snowfall and/or rainfall is driven by intense moisture transport from the Atlantic basin, towards northern part of France and western part of Germany, in narrow and long bands, which in contact with high mountain regions over the Moselle catchment area and parts of Rhine catchment area lead to extreme flooding. The precipitation anomalies associated with extreme flood peaks are not just local, but they occur on a larger spatial scale (Figure S8 and S11). The spatial structure of the precipitation reflects the mean direction with which the IVT and thus the ARs are moving towards Europe. The typical large-scale synoptic circulation pattern, leading to heavy rainfall events and extreme flooding in the lower part of Rhine's catchment area, is characterized by a deep and mobile low pressure center south of Greenland, which migrates towards the northern part of Europe, and a high pressure system over the northern part of Africa and southern part of Europe. The dipole-like structure in the SLP field leads to a south-westerly flow over France and western Germany. As the plume of moisture ascends in the warm sector of the extra-tropical cyclone and it is forced to rise over the Vosges Mountains in France, and Hunsrück and Eifel Mountains in Germany, it precipitates out, thus producing extreme rainfall in a relatively narrow band and extreme flooding some days later.

The strong pole to equator temperature gradient, in winter, results in an enhanced baroclinic zone and storm tracks affecting the western part of Europe. The extratropical cyclones, associated with the extreme flooding events over western part of Europe, including the lower catchment area of Rhine river, grow in these baroclinic zones which also contain the ARs that make landfall over the European land mass (Lavers and Villarini 2013b).

The influence of ARs on the Rhine River flood events is done via the prevailing large-scale atmospheric circulation and most of the ARs associated with these flood events are embedded in the trailing fronts of the extratropical cyclones. The evolution of the atmospheric circulation during the days prior the floods are limited mainly to changes in the amplitude of the sea level pressure. The low pressure systems develop a stronger anomaly than its high pressure counterpart. The dipole SLP pattern observed during the days prior to the flood peaks, is reminiscent of the positive phase of NAO and it guides the water vapor is a narrow band through France and the British Chanel until the western part of Germany. This was also confirmed by looking at the daily and monthly values of the NAO index during the days prior to the floods events or the monthly NAO index during the month of the flood. In 9 out of the 10 extreme floods events, the monthly NAO was in a positive phase. The influence of NAO on the flooding events over our analyzed region is made via the influence on the frequency and direction of the extratropical cyclones. The phase of NAO has a strong impact on the extratropical cyclones frequency, affecting both the location and the orientation of the cyclone tracks, extreme cyclones occurring more (less) frequently during strong positive (negative) NAO phases (Pinto et al. 2009).

The higher amplitude of the low pressure systems indicate the importance of the extratropical cyclones in directing the storm tracks towards the central part of Europe. The synoptic situation, in a PV framework, for the days characterized by heavy rainfall, exhibits meridional elongated PV anomalies, associated with anticyclonic Rossby wave breaking. ARs activity over western Europe is linked with mid-latitude Rossby wave breaking and strong PV anomalies (Zavadoff and Kirtman 2020). The flood peaks of 1925/26, 1993 and 1995 share in common the passage of sharp meridional PV gradient associated with anticyclonic RWB. These PV transitions at the tropopause level are accompanied, in all analyzed cases, by the advection of warm and humid air over Rhine catchment area and cold air intrusions over the eastern part of Europe. PV streamers and anticyclonic Rossby wave breaking have been associated also with extreme precipitation and flooding over the Alpine region (Martius et al. 2006; Froidevaux and Martius 2016; Rimbu et al. 2020).

One of the most interesting finding of this study is the fact that the extreme floods are preceded, especially 4-5 days in advance (Figure 12), by intense moisture transport from the sub-tropical Atlantic, in the form of ARs. The time lag between the AR occurrence and flood peak is related to the fact that multiple factors (e.g. duration of precipitation, time travel from the tributaries to Köln gauging station, snowpack, soil moisture) are influencing the magnitude of the flood wave. Thus, this time lag between the ARs occurrence and the flood peak, in the lower part of the catchment area of Rhine river, can be used as a potential predictor for the upcoming floods in the lower part of Rhine catchment area. Overall, the North Atlantic ARs are projected to increase both in magnitude and frequency, implying a greater risk of extreme rainfall and flooding (Lavers et al. 2013c), thus more studies are needed to test if also smaller flood peak are associated with intense moisture transport and their potential predictability.

This study adds new understanding of the meteorological processes leading to the occurrence of extreme rainfall events and flooding in the central part of Europe. Identifying ARs as a potential contributor to floods in the lower part of Rhine River catchment area, thus inland Europe, indicates the need for further studies to better understand the drivers of hydrometeorological extremes over different parts of Europe, thus allowing for a better assessment of flood risk.

**Acknowledgements.** This study was promoted by Helmholtz funding through the Polar Regions and Coasts in the Changing Earth System (PACES) program of the AWI. Funding by the AWI Strategy Fund Project - PalEX and by the Helmholtz Climate Initiative - REKLIM are gratefully acknowledged.

**Author contributions.** MI designed the study and wrote the paper. VN and BG helped with the writing of the paper and interpret the results.

**Competing interests.** The authors declare that they have no conflict of interest.

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

**Table 1**. Date of occurrence and magnitude of the ten flood events recorded at Köln gauging station used in this study.

| Date | Magnitude |
|---|---|
| 31.03.1845 | 9.800 m³/s |
| 5.02.1850 | 9.710 m³/s |
| 29.11.1882 | 10.200 m³/s |
| 16.01.1920 | 10.700 m³/s |
| 1.01.1926 | 10.900 m³/s |
| 2.01.1948 | 9.890 m³/s |
| 25.02.1970 | 9.690 m³/s |
| 29.03.1988 | 9.550 m³/s |
| 24.12.1993 | 10.600 m³/s |
| 30.01.1995 | 10.700 m³/s |







**Table 2**. Daily values of the magnitude of the IWT for the days prior to each flood peak (see Table 1) averaged over the box 4°-12°E; 47°-56°N. The shaded grey boxes represent the days when the highest magnitude over was recorded. The climatology was computed over the period 1961 – 1990.

| | 1845 | 1850 | 1882 | 1920 | 1926 | 1948 | 1970 | 1988 | 1993 | 1995 |
|---|---|---|---|---|---|---|---|---|---|---|
| **Lag 7** | 246.25 | 241.38 | 142.59 | 175.70 | 150.39 | 226.45 | 63.53 | 89.05 | 245.60 | 222.54 |
| **Lag 6** | 90.24 | 62.06 | 289.43 | 329.78 | 232.27 | **359.86** | 88.23 | 146.34 | **281.42** | **324.16** |
| **Lag 5** | 139.43 | 35.05 | **396.43** | **481.75** | 344.47 | **468.97** | **227.72** | **207.93** | **618.99** | **331.26** |
| **Lag 4** | **228.80** | **225.81** | **340.10** | **454.43** | 287.79 | **229.94** | **221.48** | **269.74** | **419.36** | **153.40** |
| **Lag 3** | **399.21** | **336.96** | **314.03** | **439.36** | **484.90** | 90.53 | **301.12** | **203.16** | 194.42 | **350.20** |
| **Lag 2** | **235.00** | **293.96** | 148.03 | 209.42 | **713.26** | 48.72 | 169.46 | 151.01 | 194.19 | 290.37 |
| **Lag 1** | 112.78 | 161.67 | 60.79 | 116.66 | **378.41** | 153.66 | 91.09 | 87.51 | 169.60 | 133.06 |
| **Lag 0** | 154.05 | 143.08 | 32.79 | 252.85 | 150.32 | 343.95 | 46.33 | 140.85 | 56.34 | 109.88 |
| | | | | | | | | | | |
| **Climatology** | **80.02** | **64.98** | **118.27** | **92.88** | **106.13** | **106.13** | **64.98** | **80.02** | **106.13** | **92.88** |








**Table 3.** Monthly values of the NAO index (second column), days with anticyclonic Rossby wave breaking (ARWB, third column) and the type of circulation patterns(GWL, fourth column) active during the days prior to each of the 10 extreme flood peaks at Koln gauging station.

| | Monthly NAO | ARWB | GWL |
|---|---|---|---|
| 31.03.1845 | -0.54 | 22 – 24.03.1845<br>28 – 29 .03.1845 | |
| 5.02.1850 | 4.13 | 31.01 – 2.02.1850<br>5 - 02.1850 | |
| 29.11.1882 | 2.01 | 23 – 26.11.1882<br>28 – 29.11.1882 | 23 – 25.11.1882 -» WZ<br>26 – 29.11.1882 -» NWZ |
| 16.01.1920 | 2.84 | 10 – 15.01.1920 | 10 – 14.01 -» WZ<br>15 – 16.01 -» WA |
| 1.01.1926 | 0.29 | 27 – 31.12.1925 | 18 – 30.12 -»WS<br>31.12 – 1.01 -»WZ |
| 2.01.1948 | 0 | 25 – 28.12.1947 | 26 – 30.12 -» WS<br>31.12 – 2.01 -» WZ |
| 25.02.1970 | 1.10 | 19 – 23.02.1970 | 18 – 20.02 -» WZ<br>21 – 24.02 -» WW |
| 29.03.1988 | 0.78 | 23 – 27.03.1988 | 21 – 28.03 -» WS |
| 24.12.1993 | 2.17 | 18 – 24.12.1993 | 8 – 24.12 -» WZ |
| 30.01.1995 | 2.70 | 26 – 30.01.1995 | 22 – 30.01 -» WZ |




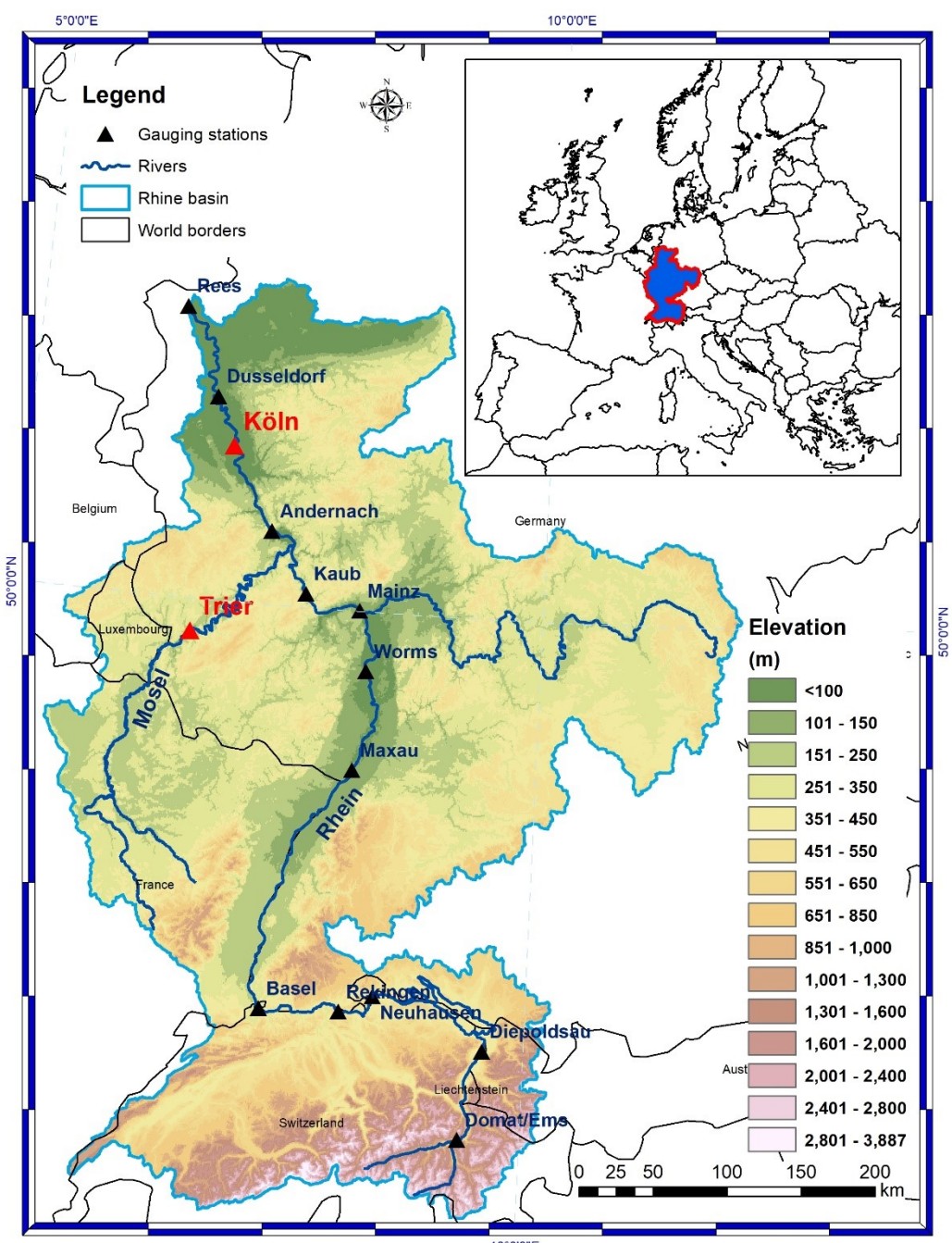

**Figure 1.** Rhine River catchment area (black contour) and the location of Trier meteorological station and Köln gauging station. The digital elevation model data was extracted from the WorldClim 2 dataset (Fick and Hijmans 2017).



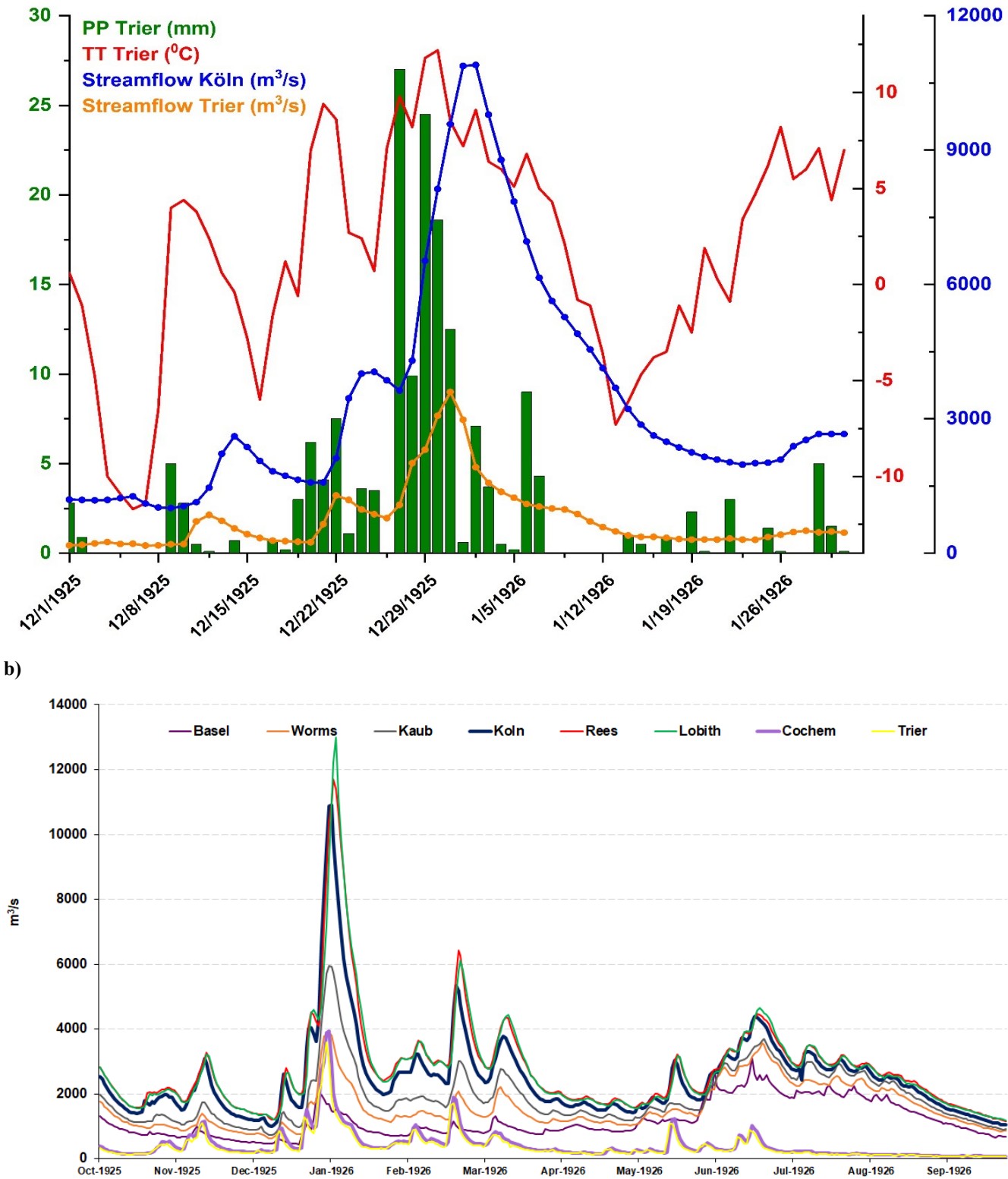

**Figure 2**. a) Daily precipitation (green bars) at Trier meteorological station, daily mean temperature (red line), daily streamflow at Köln gauging station (blue line) and daily streamflow at Trier gauging station (orange line) for the period 1.12.1925 – 31.1.1926 and b) Daily streamflow at different gauging station along Rhine River (Basel, Worms, Kaub, Köln, Rees, Lobith) and Moselle River (Trier and Cochem) for the period 1.10.1925 – 30.09.1926.

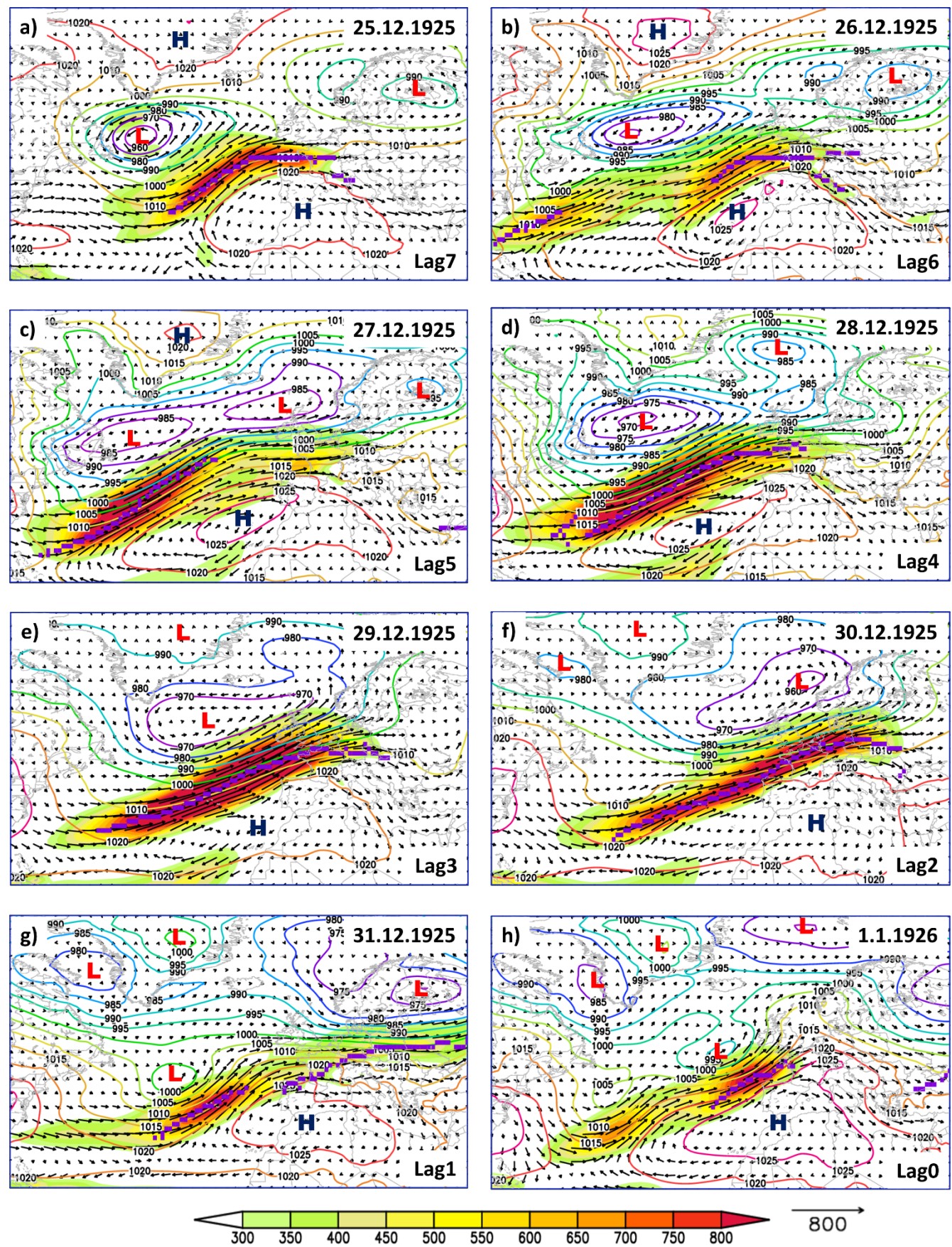

**Figure 3.** Daily sea level pressure (SLP, colored contour lines), magnitude of the integrated water vapor transport (IVT, shaded colors), direction of the integrated water vapor transport (vectors) and location of the AR axis (magenta line) for different time lags (0 – 7 days) for the 1925/26 flood event. Units: SLP (hPa) and IVT (kg·s$^{-1}$·m$^{-1}$).

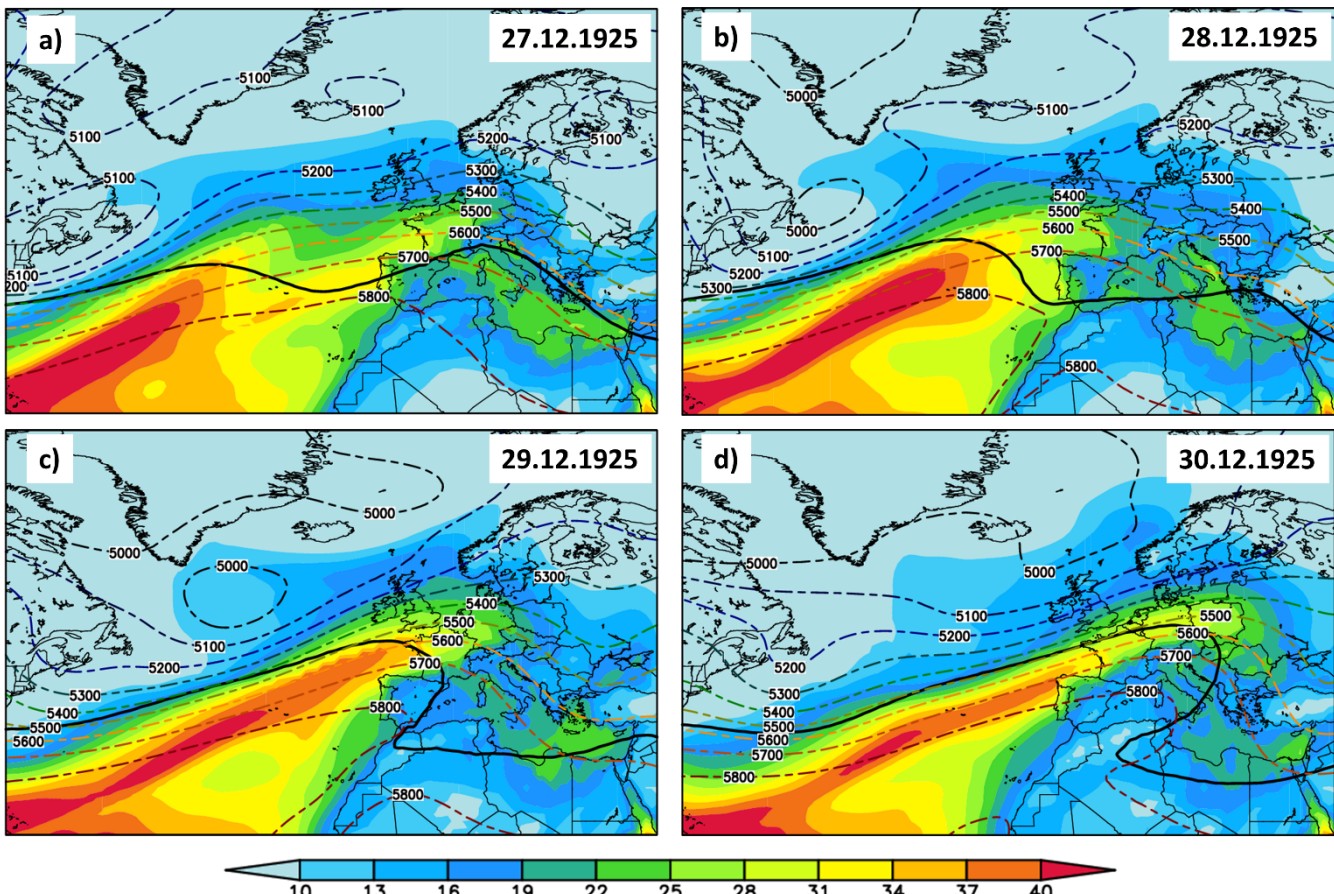

**Figure 4.** Daily integrated water vapor (IWV, shaded colors) and daily geopotential height at 500 hPa (Z500, contour lines) for a) 27.12.1925; b) 28.12.1925; c) 29.12.1925 and d) 30.12.1925. The thick black line in a) – d) indicates the 2PVU contour at 330K. Units: IWV (kg m$^{-2}$) and Z500 (m).





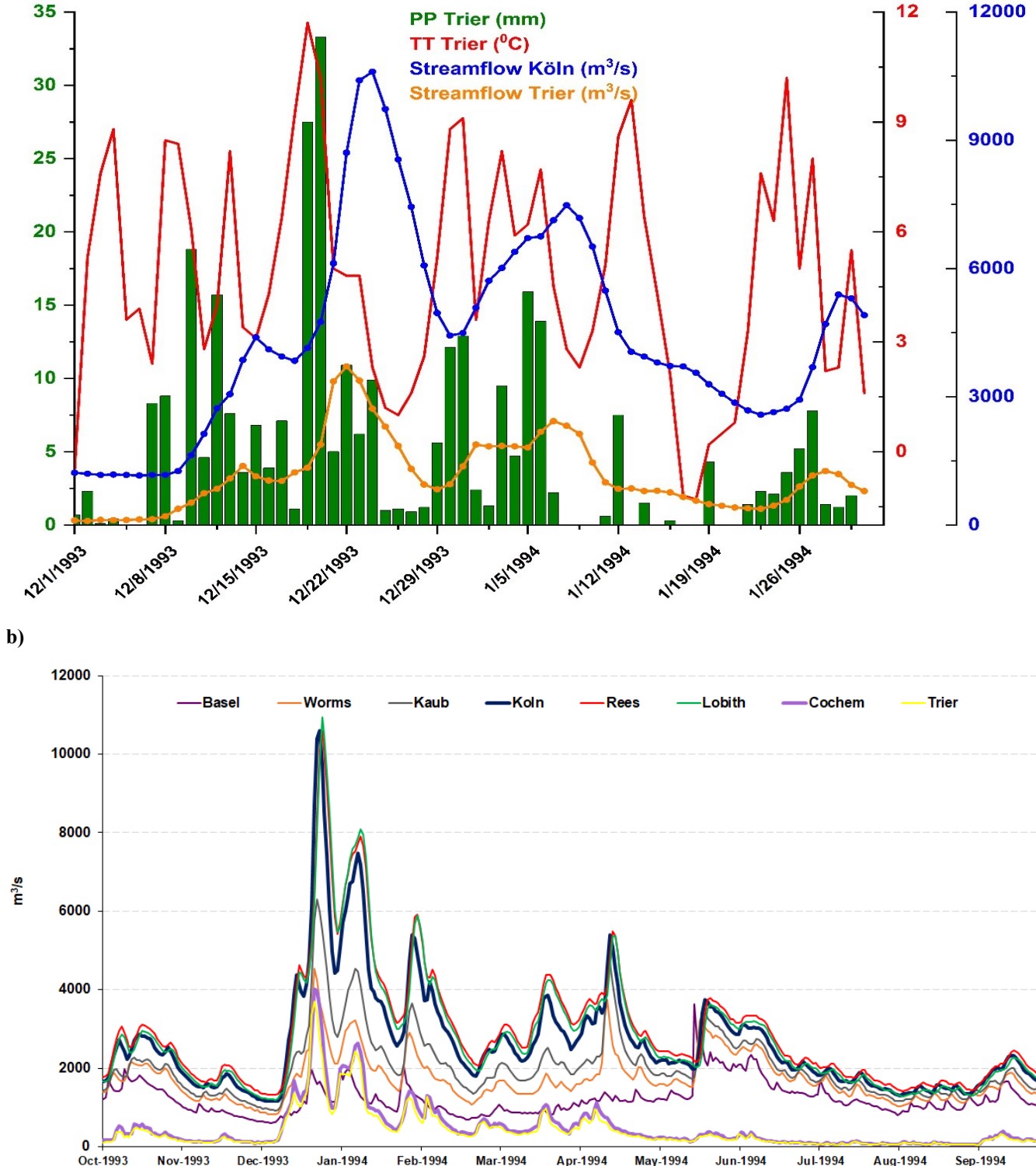

**Figure 5.** a) Daily precipitation (green bars) at Trier meteorological station, daily mean temperature (red line), daily streamflow at Köln gauging station (blue line) and daily streamflow at Trier gauging station (orange line) for the period 1.12.1993 – 31.1.1993 and b) Daily streamflow at different gauging station along Rhine River (Basel, Worms, Kaub, Köln, Rees, Lobith) and Moselle River (Trier and Cochem) for the period 1.10.1993 – 30.09.1994.


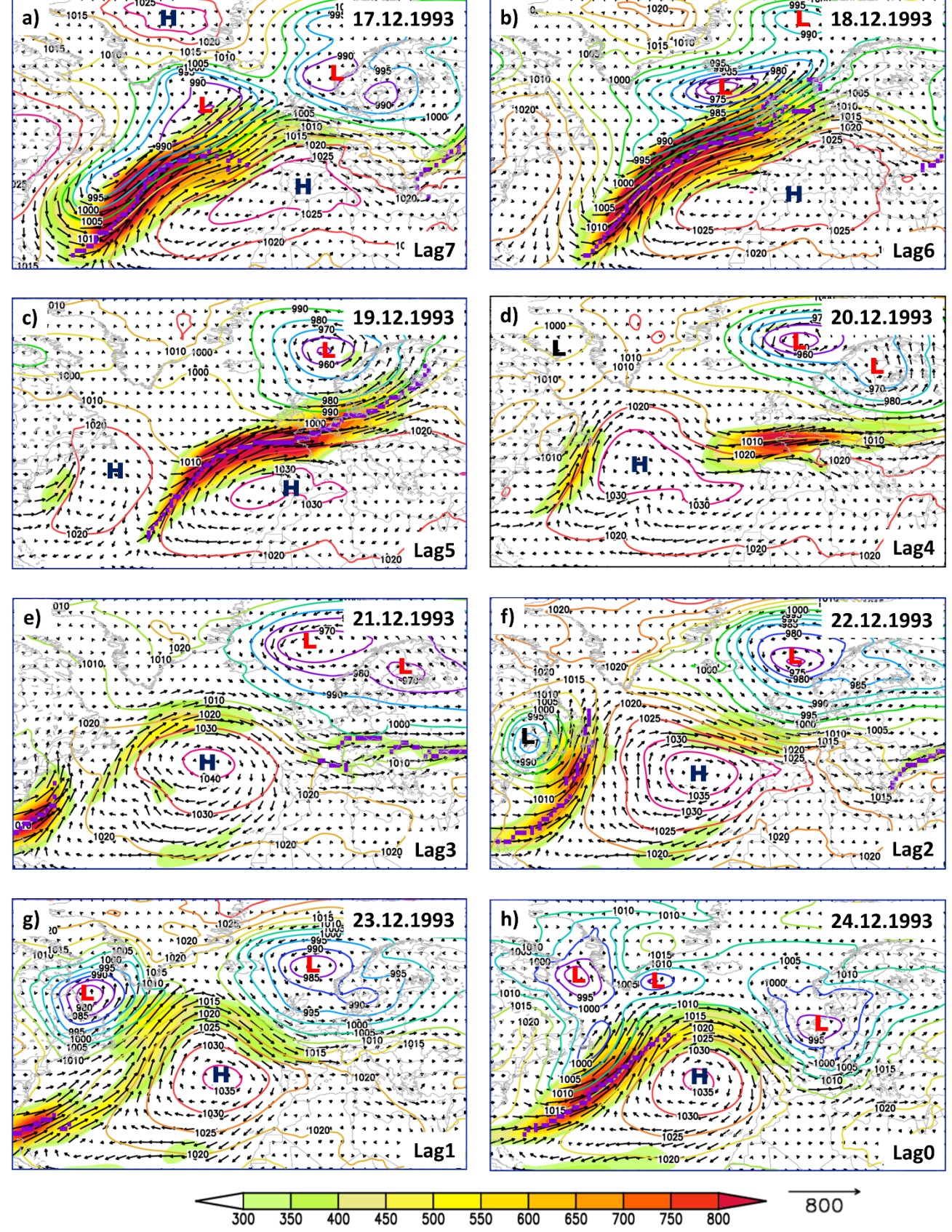

**Figure 6.** Daily sea level pressure (SLP, colored contour lines), magnitude of the integrated water vapor transport (IVT, shaded colors), direction of the integrated water vapor transport (vectors) and location of the AR axis (magenta line) for different time lags (0 – 7 days) for the 1993 flood event. Units: SLP (hPa) and IVT (kg·s$^{-1}$·m$^{-1}$).

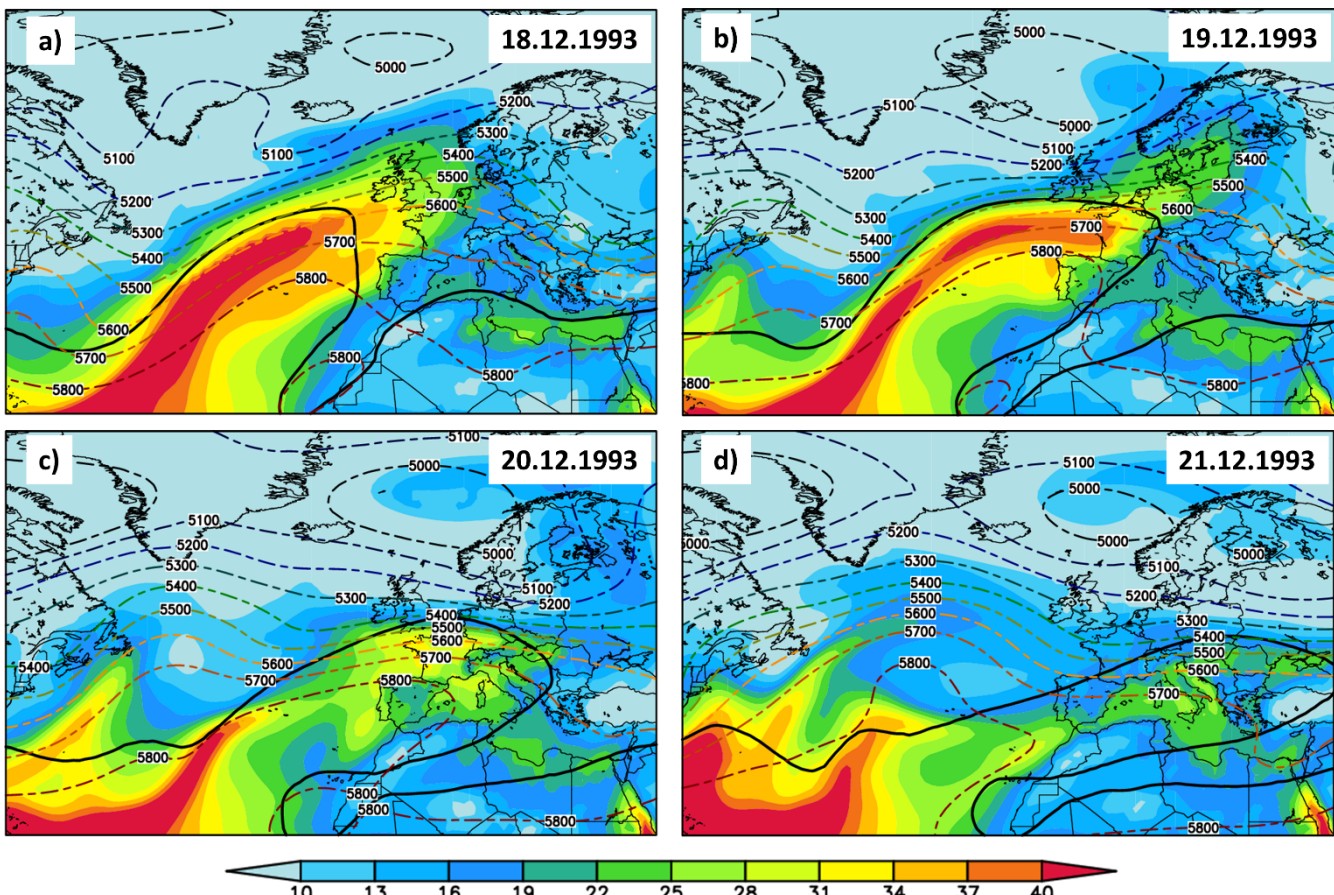

**Figure 7.** Daily integrated water vapor (IWV, shaded colors) and daily geopotential height at 500 hPa (Z500, contour lines) for a) 18.12.1993; b) 19.12.1993; c) 20.12.1993 and d) 21.12.1993. The thick black line in a) – d) indicates the 2PVU contour at 330K. Units: IWV (kg m$^{-2}$) and Z500 (m).





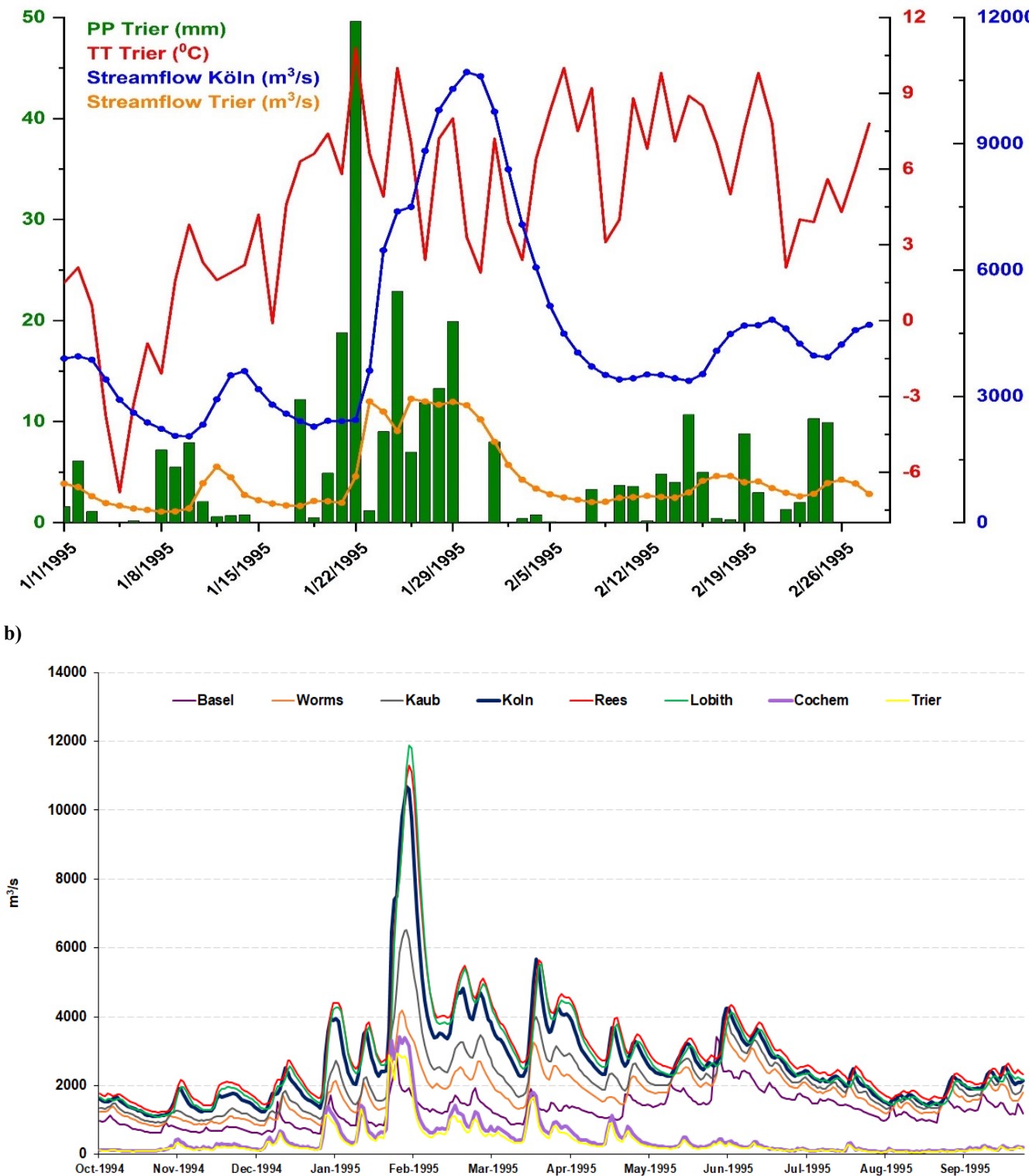

**Figure 8.** a) Daily precipitation (green bars) at Trier meteorological station, daily mean temperature (red line), daily streamflow at Köln gauging station (blue line) and daily streamflow at Trier gauging station (orange line) for the period 1.1.1995 – 28.2.1995 and b) Daily streamflow at different gauging station along Rhine River (Basel, Worms, Kaub, Köln, Rees, Lobith) and Moselle River (Trier and Cochem) for the period 1.10.1994 – 30.09.1995.


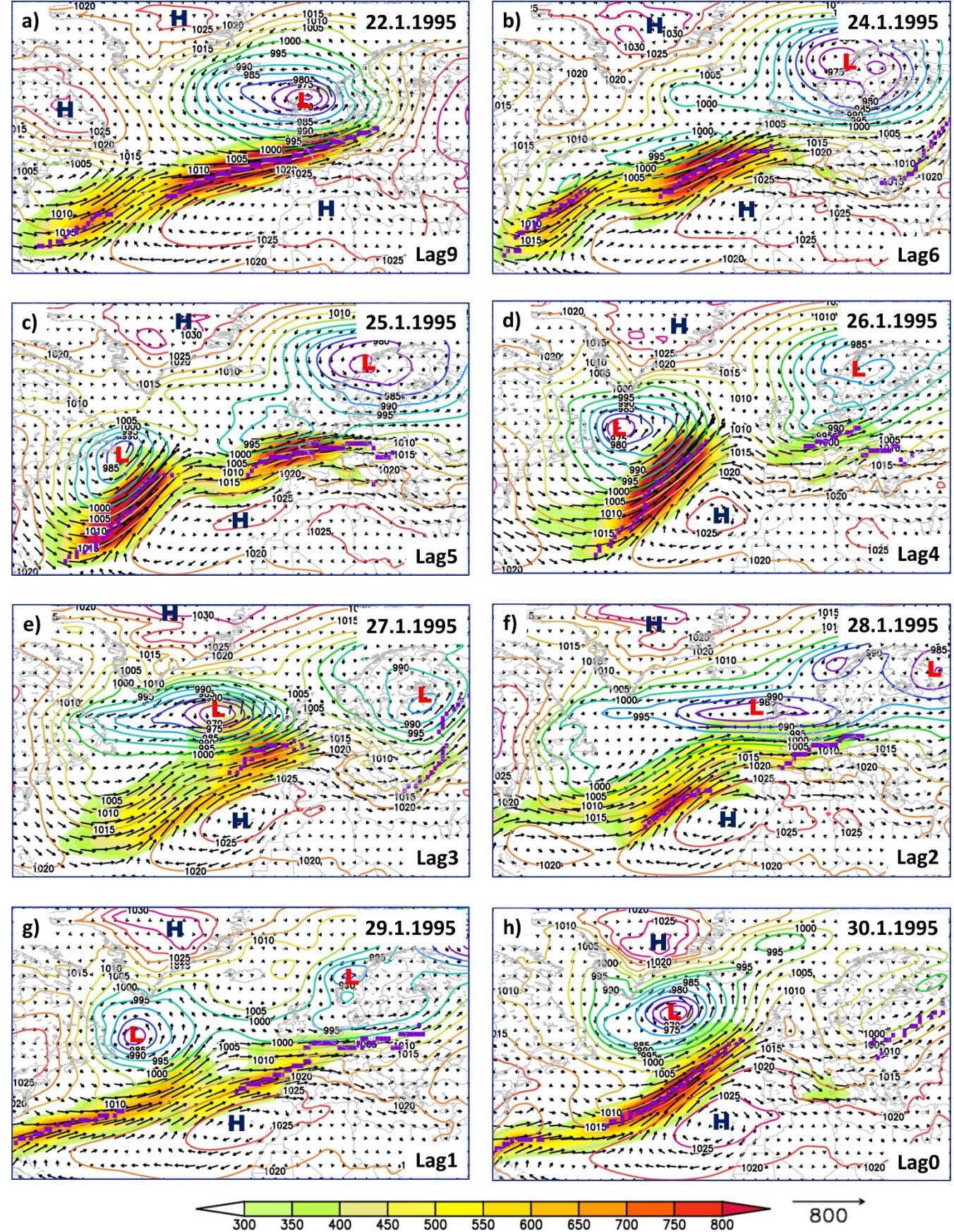

**Figure 9.** Daily sea level pressure (SLP, colored contour lines), magnitude of the integrated water vapor transport (IVT, shaded colors), direction of the integrated water vapor transport (vectors) and location of the AR axis (magenta line) for different time lags (0 – 7 days) for the 1995 flood event. Units: SLP (hPa) and IVT (kg·s$^{-1}$·m$^{-1}$).

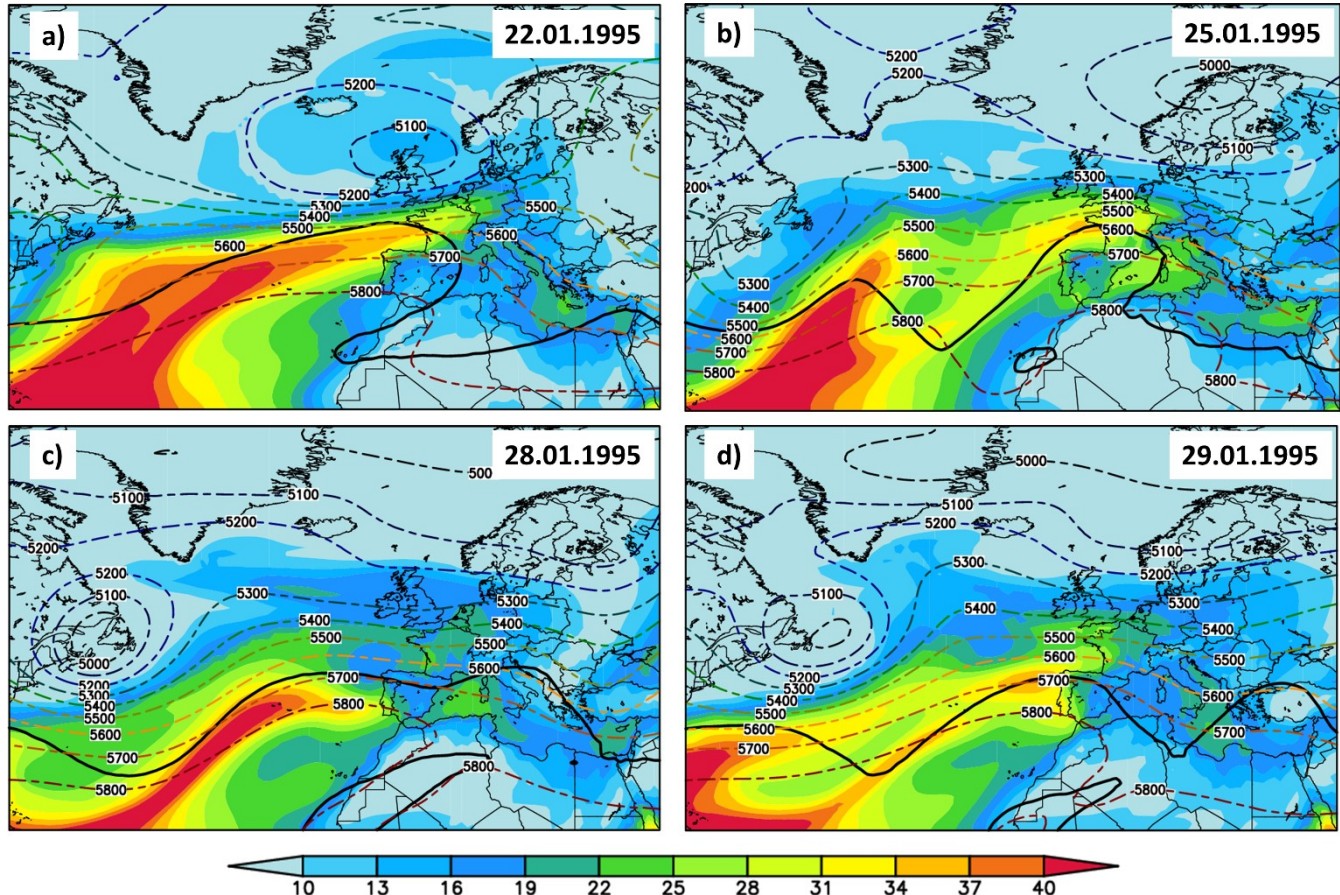

**Figure 10.** Daily integrated water vapor (IWV, shaded colors) and daily geopotential height at 500 hPa (Z500, contour lines) for a) 22.01.1995; b) 25.01.1995; c) 28.01.1995 and d) 29.01.1995. The thick black line in a) – d) indicates the 2PVU contour at 330K. Units: IWV (kg m$^{-2}$) and Z500 (m).


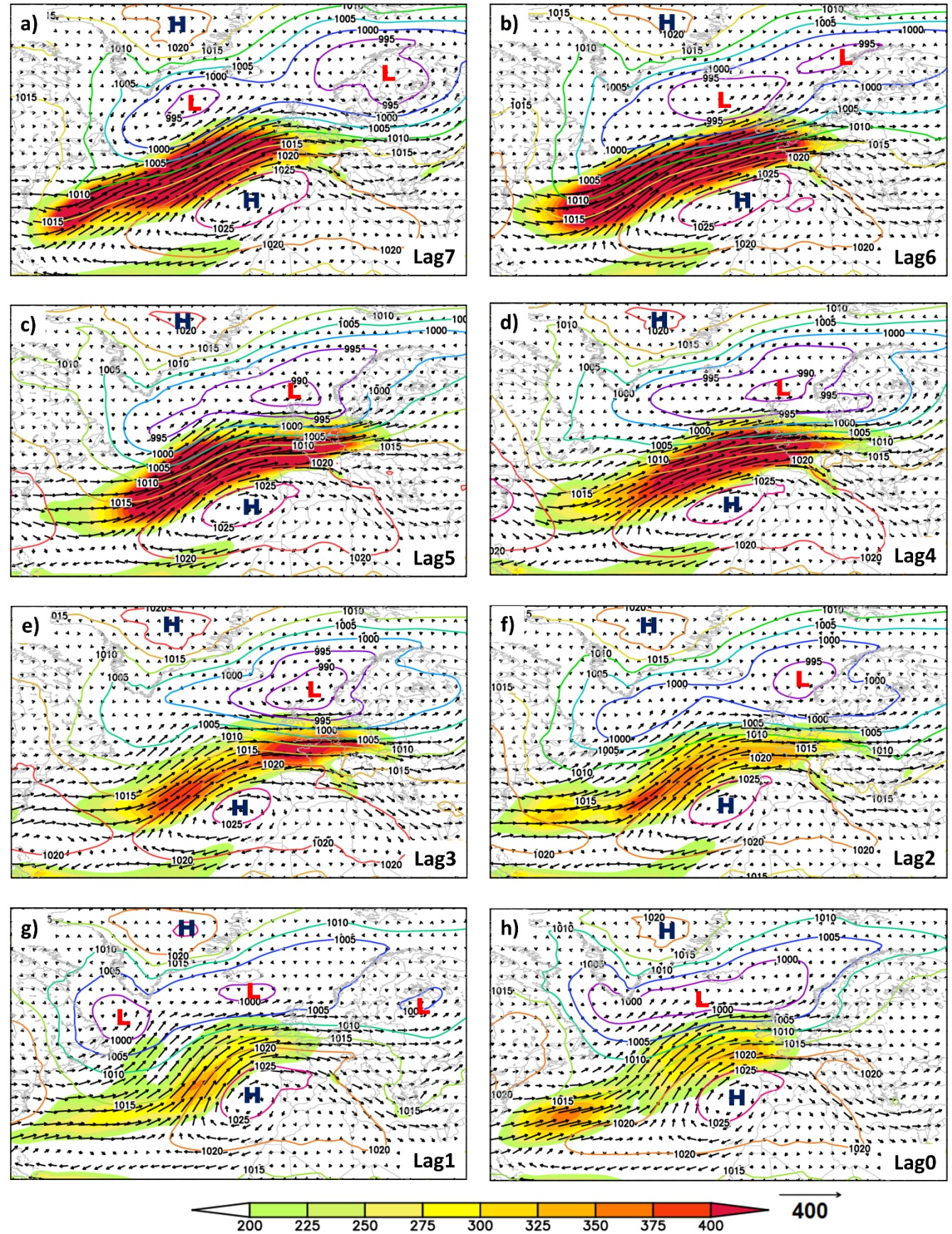

**Figure 11**. Composite of the mean sea level pressure (SLP, colored contour lines), magnitude of the integrated water vapor transport (IVT, shaded colors) and the direction of the integrated water vapor transport (vectors) for different time lags (0 – 7 days) for the 10 highest flood peaks recorded at Köln gauging station (see Table 1). Units: SLP (hPa) and IVT (kg·s$^{-1}$·m$^{-1}$).

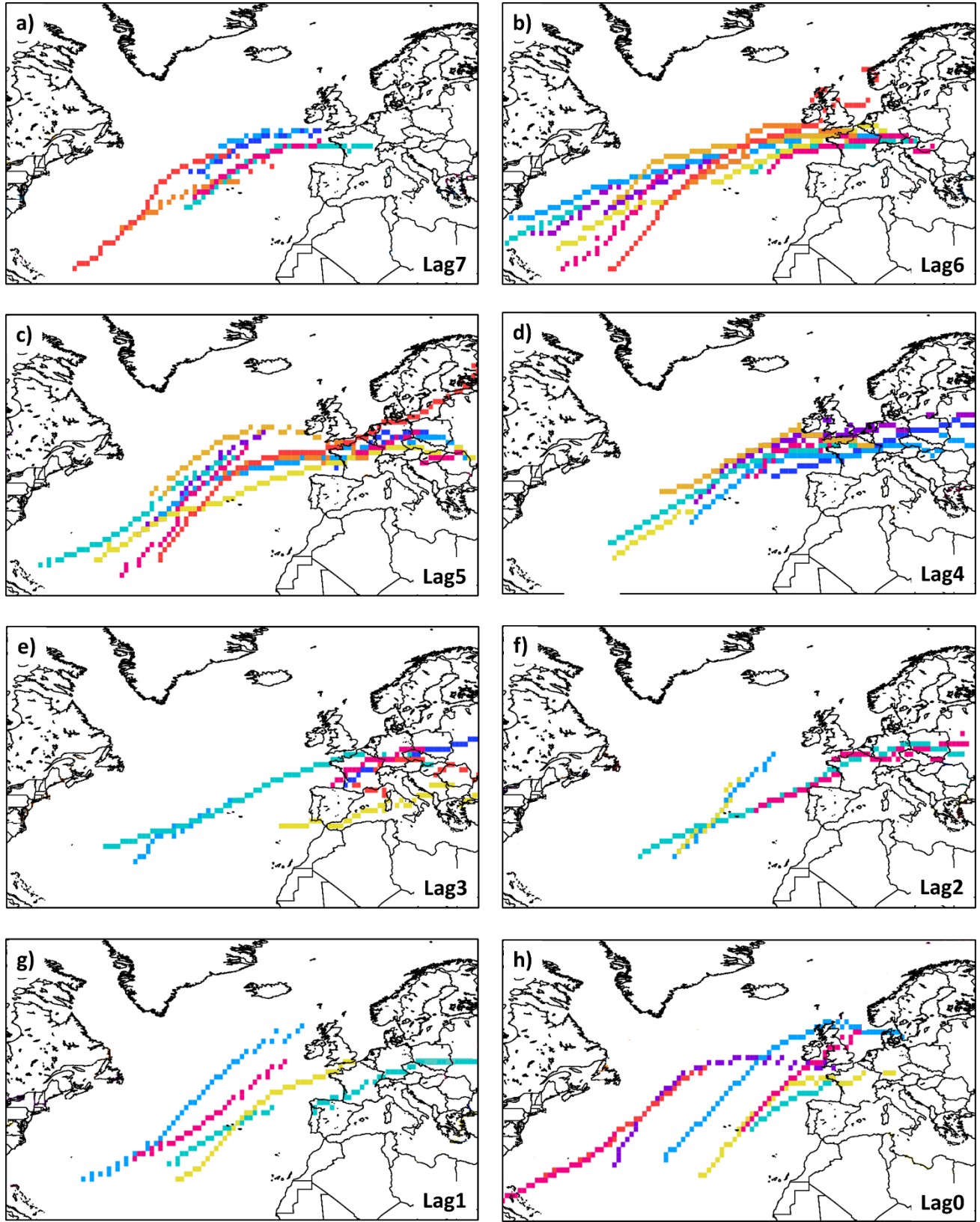

**Figure 12.** The AR axis location for the top 10 winter floods with different time lags (0 – 7 days). Each color is assigned to a flood peak.

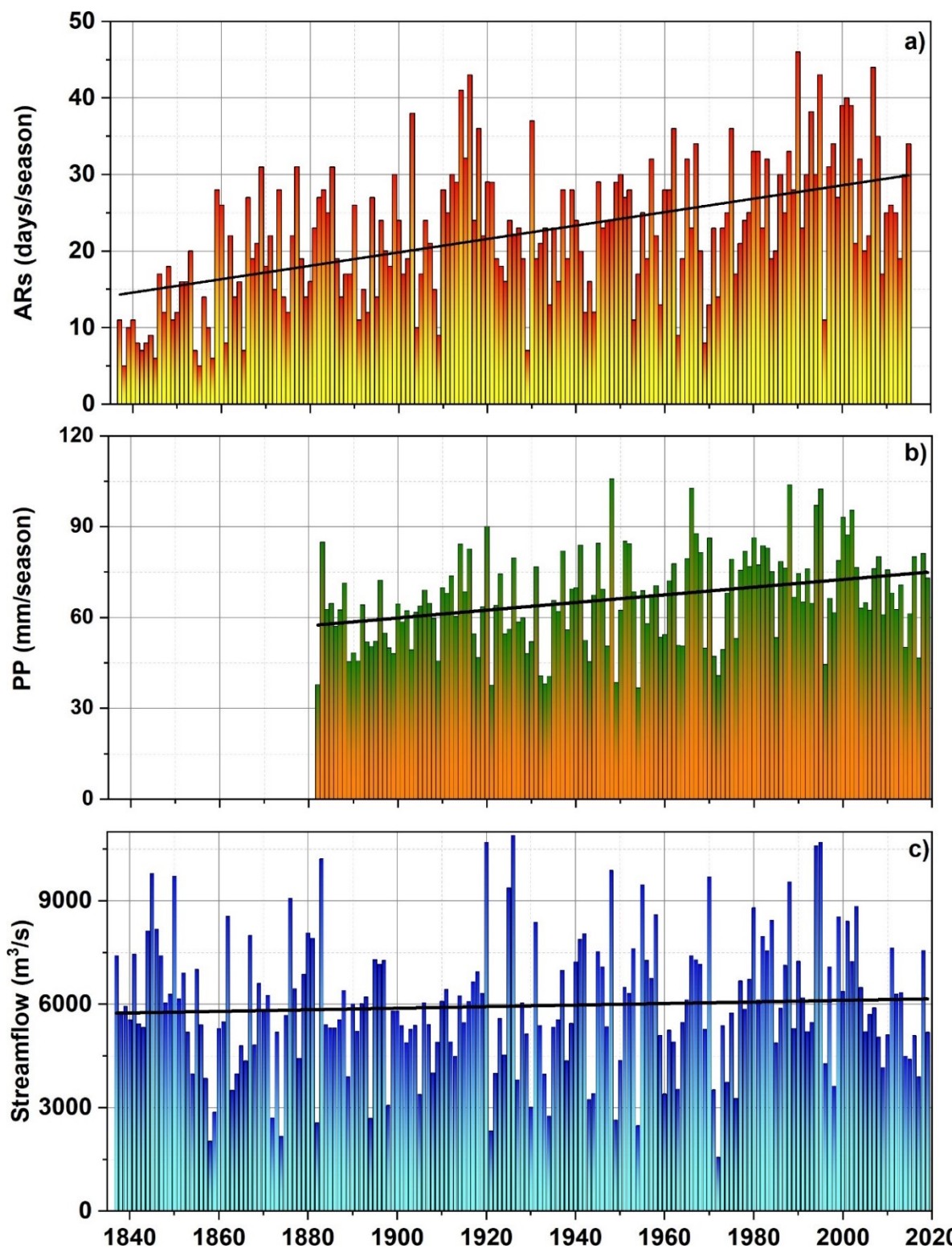

**Figure 13**. a) Number of days/season with ARs over the region 4° - 12°E; 47 – 56°N, for the period 1836 – 2015; b) Seasonal precipitation averaged over the German side of Rhine catchment area for the period 1881 – 2019 and c) Daily maximum streamflow at Köln gauging station over the period 1836 – 2019. For a), b) and c) we used only the period November – March for the analysis.