# Peer review of "Rivers in the sky, flooding on the ground: the role of atmospheric rivers for the inland flooding in central Europe"

_Hydrology and Earth System Sciences, 2020_

## Referee Comment (RC1) · Anonymous Referee #1 · 3 May 2020

The paper provides an interesting and unique contribution showing atmospheric rivers as drivers of high flows in the lower Rhine catchment. It thus enriches our understanding of hydrometeorological flooding drivers at the catchment level for this global region. By utilizing a long and comprehensive meteorological record, the authors show how indeed ARs have led to important damages in the region. It also provides interesting insights these events are preceded up to 7 days by intense moisture transport from the tropical North Atlantic basin typically precede ARs. The comments I have are minor which concern mainly methodological clarifications as well as suggestions to provide more insights of the repercussions of their findings. If a new version of the manuscript successfully address these issues I would recommend the article for publication. Please find here below my specific observations:

- Line 78: "i) analyze from a hydrological point of view" This sounds rather broad and ambitious; this line should be refined to specify what 'hydrological' actually means in the context of this publication
- Line 81: "iii) link the flood peaks with 80 the occurrence of AR". It does not look very scientifically sound to try to find a link between these two aspects. This suggest that the objective is already anticipating the results. This objective should be changed to something along the lines of 'explore how the occurrence of ARs explains flood peaks' or similar
- 95-102: This paragraph should be enriched with references. It sounds as if data has been already processed and the authors are already presenting their results
- Line 98: "winter or spring floods, which are triggered by warm air intrusions with corresponding snow melt in flatlands and low mountain ranges and summer floods, which are fed by large-scale 100 heavy rain or long-lasting repeated precipitation episodes (in connection with late snowmelt / glacier runoff in theAlps)" Please improve the fluidity of this sentence or try to split it two. At present it is not very clear
- Line 112: This paragraph would be enriched by a short 1-2 sentence conclusion about the general hydrological trends of these tributaries.
- Line 114: How is this station representative for the catchment? What about the impact of upstream water infrastructure (dams, reservoirs, levees, and others which may mitigate floods)? Wouldn't readings at this specific part of the catchment give a misleading interpretation if the catchment is heavily intervened? Is hydraulic and infrastructure intervention indeed important here?. I think that this is very relevant when you compare events of 1925 vs the 1990s. I suppose that a number of hydraulic interventions have been made in the river in a period of ~60 years. Probably these interventions and the fact that you are just looking at river gauge readings are leading us to underestimate the connection between ARs and flood peaks. Even if the role of hydraulic infrastructure is not deemed as relevant in this part of the catchment, it would be useful to clarify these aspects throughout the text.
- Line 144: reference? What is the general proportion of floods happening during winter vs summer? Are winter ones more or less frequent (generally speaking although these trends, naturally, are not constant)?
- Line 214 and in general throughout the text: The manuscript would be highly enriched if somehow these monetary losses are translated to current usd/eur

values; not asking the authors to perform a complex econometric calculation but it would be useful to put these economic losses in perspective. For example, the 1925 event seemed to have caused losses of 100 Million DM vs 50 Million DM in 1993 vs 500M is 1995. How do they compare each other nowadays in current USD/EUR?. Similarly, the text would be enriched if the authors provide a table (or figure) comparing the impacts that these events have had on human lives, displaced people, monetary losses, infrastructure damages (even a qualitative description), and others. This would provide a useful information to understand the truly impacts of ARs in this key global catchment.

- Line 333, Conclusions:
  - In general I think that either here or in previous sections there should be a short discussion describing the general trends of ARs-caused high flows over these 2-century time period. With the data you already have, it would be very useful to have a perspective on whether ARs-caused floods in the Rhine have been more recurrent? Or more intense? Both? None? While I understand that a full and comprehensive trend type of analyses might be out of the scope of this study, the manuscript will be highly enriched if even few sentences are added exploring this issue.
  - The conclusions should also highlight the socio-economic impacts that these events have caused.

---

## Referee Comment (RC2) · Anonymous Referee #2 · 12 May 2020

**Review: Rivers in the sky, flooding on the ground**

This paper investigates the relationship between atmospheric rivers and extreme flooding in the lower Rhine basin. So far, studies mostly investigated this relation between extreme rainfall (or flooding) and ARs for coastal regions. Therefore, the objectives to analyze the connection between ARS and floods occurring more inland is relevant and interesting, and fits the scope of HESS. The study describes the hydrometeorological situation of the three most extreme flood events over the lower Rhine basin in the last 180 years, and the connection with atmospheric rivers. In addition, composites of the large-scale circulation and IVT describing atmospheric rivers are analysed for the 10 most extreme flood events.

Although the analyses answer the research objectives, in my view there is much more potential and knowledge to gain from the dataset and method then is done so far in the manuscript. In fact, the section on the composites of the 10 largest flood events is very short, and has much more potential. I will give a few suggestions on additional questions/ experiments, and leave it up to the author/editor if these analyses are needed in this manuscript or can be potentially used as research questions for further studies.

In addition, I have some major comments and unclarities on the paper below, which should be addressed before publishing. Furthermore, the writing of the manuscript can be improved, being consistent in tense, English language, and use of units throughout the manuscript. Please find my minor comments at the end of this document.

Additional questions arising from the manuscript:

- How are the trends in flooding and ARs over the dataset? The dataset spans quite a substantial time (1836-2015) and in the method section it is indicated that floods probably would happen more frequently (Line 101-110), so it would be nice to explore this further using this dataset. You can also assess if Atmospheric Rivers are going to be more frequent and intense over the lower Rhine catchment as you refer to that is the case over the North Atlantic (line 370)
- As Atmospheric Rivers are associated with the Warm Conveyor Belt of extratropical cyclones, besides holding a lot of moisture, temperatures are often warmer than normal within these ARs. The effect of temperature on snow melt in the Alps or lower Rhine basin could positively influence discharge peaks. This aspect remains underexposed in this study. For example in the case of high temperature inducing snow melt could be related.
- I miss the explanation of the mechanism how Atmospheric Rivers result in extreme rainfall (for coastal areas; when an AR reaches a topographical barrier air parcels are lifted and adiabatically cooled, resulting in clouds and precipitation). For the lower Rhine this lifting mechanism is probably related to the Ardennes area? I am wondering if these mountains trigger enough uplift to result in precipitation or that the amounts of moisture during the three investigated cases is so high that only little uplift is needed.
- You have selected the 3/10 most extreme flood events in the lower river Rhine area and linked those to Atmospheric Rivers. In a dataset of 180 years, more flood events could be selected and their link with Atmospheric Rivers can be investigated.

With that, the robustness of the link between ARs and extreme rainfall and flooding over the lower Rhine basin can be analyzed and put into perspective

**Major comments:**

The title is illustrative, although does not exactly reflect the novelty of this research with the focus on the link between large-scale circulation (ARs) and floods **inland**.

Atmospheric studies such as Helen Dacre (2015) argue that atmospheric rivers are a result of constant local recycling of water along the cold front of an extratropical cyclone. This suggests that precipitation related to ARs originates more locally rather than from sub-tropical regions as is indicated in line 68, 74 , 336 etc. Please discuss or revise.

In the introduction ARs and the relation with different teleconnections is mentioned, but I don't see that coming back in the rest of the paper. Have you checked NAO indices for the events you studied? This could be interesting, as you often refer to a low south of Greenland (Iceland) and a high over the coast of North Africa (Azores), which gives indication for a positive NAO resulting in strong flows to northwestern Europe. It would be interesting to invest the index of NAO for the selected flood events.

Although your showing IVT in the figures, I miss the embedding in the text. The case studies could have more focus on this IVT values and how anomalous they are, as to my knowledge values in IVT of 800 kg/m/s are quite exceptional. It would be nice to put this a bit more in perspective

The lag between AR/extreme precipitation is interesting and could deserve some more attention throughout the manuscript. Of course this mainly depends on local timing and location, indicating the importance of local processes (to be added on line 40) although it depends on the size of your catchment.

Both in the conclusion as in the abstract I miss quantification of the results. Can you give some numbers to align your statements? For example indicate how anomalous the selected events were in terms of discharge and IVT related to the AR. I think an important conclusion from this research is that in these extreme events, large-scale circulation are rather similar but local conditions leading to the flood not. This is an interesting conclusion which could be highlighted more, and could be strengthened if it was further quantified in the conclusion and abstract as well.

**Minor comments:**

Line 8: The role of  large scale atmospheric circulation

Line 13: sentence: The influence of ... In my view atmospheric rivers are part of the large-scale circulation, the .. is done via the prevailing large-scale atmospheric circulation.. sounds very odd to me

Line 34: that coping with floods is not **trivial**

Line 40: I would argue that for flood forecasting and the time of occurrence, location and magnitude scales from mesoscale to local scale are important. Especially knowledge on local orography is needed to get the location of the flooding right

Line 42: Same sentence as last sentence of previous paragraph.. combine?

Line 49: This sentence is not clear. What do you mean with regional and local climates?

Line 115: Rhein --> Rhine

Line 118: what is the time resolution of the reanalysis data?

Line 119: I am confused that a re-analysis dataset has ensemble members, could you explain that?

Line 121: has several improvements > vague statement, either name improvements or leave out

Line 133: divided by gravity ($g$).

Lines around 133: What is the vertical resolution of your wind and specific humidity data?

Line 152: Where do you show the EOBS gridded precipitation dataset? Not clear how and if this comes back in your results. Although it would be good to analyse a gridded precipitation field over the lower Rhine basin instead of a point observation at Trier.

Line 161: .. in  large parts of the Rhine catchment..

Line 164: od -> of

Line 167: Can you quantify the total amount of precipitation over the lower Rhine basin area instead of showing the gridpoint values in the appendix. That would give more quantification to the results.

Line 169: This sounds like a positive North Atlantic Oscillation. Would be good to check the index for the selected events.

Lines around 170: Miss numbers of IVT in 1925 case, that would give indication of the 'severity' of AR. Can you give some IVT values here to give some indication as you do for the hydrometeorological situation of 1993 (line 229). In general it would be good to embed the values of IVT a bit more within the text as I think they are quite exceptional, can you compare with climatology? Or average value of ARs at this latitude (as the value of IVT is latitude dependent)

Line 180: specific humidity or moisture

Line 183 etc: Why do you talk about wind and moisture separately here? You can refer to IVT and Figure 3 and in my opinion there is no need in showing figure 4 as it gives the same information as Figure 3.

Line 223: too should be to

Line 228: Again, could you give precipitation values averaged over the basin? And compare that to climatology?

Line 238: Where is statement based on? Add reference

Line 268: become > became. Keep tenses consistent throughout the result section.

Line 268: Where is Berus meteorological station located? Indicate in map

Line 298: driver > driven

Line 319: plum should be plume?

Line 321: by a south-westerly wind (Figure 11) > you are showing IVT vectors and no wind vectors in Figure 11 so wrong reference

Line 322: By visual inspection.. etc. Are you referring to the individual ARs per event or the composites here? If you refer to the composites I would not expect to see individual ARs as the composite gives an average and the ARS are therefore smoothed and you can expect IVT with widths bigger than 1000 km wide.

Line 336: Sentence 2 in the conclusion is not a conclusion of your work, but new information: I would move it to the introduction

Line 346: what is meant by westerly, southwesterly and north-westerly large-scale circulation types? Do you mean prevailing winds? Please clarify.

Line 358-363: This sentence is an explanation why more storms (ARs) occur in winter and should be moved to the methodology section to explain why this study focuses on wintertime.

Line 363: .. is done .. this sentence is not very easy to read

Line 379: I guess this research is about the UK? Maybe good to add that in the sentence to be more specific

**Table & Figures**

- I see in Table 1 that the magnitude of the flood of 1995 is as high as the one in 1920, so why was the case of 1995 described and not the one from 1920? Or is it because these stream flows are from Koln while you based your analysis on the ones from Trier? This should be stated more clearly in the text, as it is not clear from the methods.
- Also streamflow in Trier station is mentioned in Figure 4 and similar figures, while I understand from the Composite events section that you base your analysis on flood peaks measured at Koln (should be Cologne) gauging station. This is confusing.
- Missing units in Figure 3, 4 and all similar figures
- In my opinion the figures with daily specific humidity and wind at 900 hPa do not add enough additional information to be shown in the main manuscript.

- Figure 2: alone should be along
- Figure 2b and all subsequent figures. Is it needed to show daily streamflow from all these locations and for the whole year? If so, those locations should also be located in Figure 1 or mentioned in the methods section. In my opinion the a figures with discharge at Trier and Cologne give enough information and I would rather show the spatial distribution of precipitation as an addition.
- Figure 12: Not sure what the colors present here? Are these the colours for the 10 different extreme events? I cannot imagine that the orange AR just south of Greenland at Lag0 leaded to a flooding in the Alps, can you comment? This figure needs more explanation in the caption and also a color scale.

Some figures in the additional material miss units and the data sources should be clarified in more detail, are these gridded observation data, re-analysis?

**References**

Dacre, H.F., Clark, P.A., Martinez-Alvarado, O., Stringer, M.A. and Lavers, D.A., 2015. How do atmospheric rivers form?. *Bulletin of the American Meteorological Society*, *96*(8), pp.1243-1255.

I just wanted to refer to this article which just appeared in Journal of Hydrometeorology which also makes the connection between Atmospheric Rivers and floodings, but then for western Norway:

Hegdahl, T.J., Engeland, K., Müller, M. and Sillmann, J., 2020. An event-based approach to explore selected present and future Atmospheric River induced floods in western Norway. *Journal of Hydrometeorology*, (2020). https://journals.ametsoc.org/doi/10.1175/JHM-D-19-0071.1

---

## Referee Comment (RC3) · Anonymous Referee #3 · 13 May 2020

**Rivers in the Sky, flooding on the ground**

**Reviewer 3 Report Round 1**

This article analyzes the role played by atmospheric rivers in some of the most important flood events in the lower part of the Rhine River basin. Overall, the paper is well written, and the inclusion of the perspective of the hydrological extremes –floods– rather than the simple extreme precipitation is always an added value. The authors find most of the more important flood events over the region were preceded by an AR event, and this is an interesting result that could be valuable for the region. The quality of the figures is acceptable, but it could be improved. I suggest the authors improve some of them if that is not very problematic.

I believe that the title is a bit pretentious. It is a very catching title that would be probably the best choice for a review paper or a paper intended to get conclusions on a global scale. This manuscript is focused on a very particular region of Inland Europe, and I think that this should be reflected in the title somehow. I would perfectly understand if the authors would like to keep the "Rivers in the sky, flooding on the ground" –I would have done the same–, but I suggest that this title should be extended with a citation to the region of interest somehow.

I have already read the comments made by the other reviewers, and I mostly agree with them. Reviewer 2 suggests to extend the 10-events composites. I will not put that condition as necessary to give my full recommendation to publish, but I think that it could be a good improvement for the paper if the authors are willing to do it. Also, this colleague suggests the authors include a discussion about Helen Dacre's (and others) perspective of the importance of local convergence of moisture in ARs development. He/She is right, but I would like the authors to take into account –when they discuss this point– that there is also a huge bunch of articles of all kinds pointing out to the essential role played by the large scale advection of tropical and subtropical moisture. I do not think that the authors should take sides with any of those perspectives –actually, I believe that both mechanisms are necessary, and the relative importance between them changes among the events–, but both may be included in the discussion.

I will not suggest major changes, however, some of my comments (particularly those regarding the very likely explosive nature of some of the involved cyclones and also those regarding the role played by NAO) will take some time from the authors to be replied. I would like to read and discuss the answers in an eventual second round of the review process.

**Minor Comments**

**L.43** I suggest the authors consistently arrange the citations by chronological order. It is not only fairer for our colleagues, but also the result is more elegant. For example, in this case, I would start from Lavers and finish by DeFlorio or Guan and Waliser.

**L.54** Please, leave a blanck space between "50" and "km".

**L.57** The beneficial aspects of ARs are not restricted to arid/semiarid areas at all. Most ARs are beneficial even in mid-latitudes. This idea is well discussed in Ralph (2019), and I think that should be included somehow in the text.

Ralph, F. M., Rutz, J. J., Cordeira, J. M., Dettinger, M., Anderson, M., Reynolds, D., ... & Smallcomb, C. (2019). A scale to characterize the strength and impacts of atmospheric rivers. Bulletin of the American Meteorological Society, 100(2), 269-289.

**L.65** There are some other important analyses relating ARs and extreme precipitation and floods in Europe that the authors did not take into account. (e.g. Eiras-Barca et al.; 2016, 2017).

**L.87** I think that section 2 must be included in the methods section. However, this is just my opinion and I let this decision to the authors.

**L.116** If the authors had SLP, why did they start the vertical integration at the level of 1000 hPa instead of SLP, which would the most correct option?

**L.130-134** I don't see the need to describe with words what the equations are already saying.

**L.135** The algorithm (and database) developed by Guan at UCLA is one of the most commonly used in our field, and I am not going to call it into question. However, did the author consider the possibility that the detection thresholds could have substantially changed in these almost 200 years that they are considering in the analysis?

**L.136** Please, replace the asterisks by ·

**L.152** How well is performing EOBS over Germany? Some analyses pointed out the fact that EOBS may not be the best option over continental Europe...

**L.164** Please, leave a blank space between 27 and mm. Take this into account throughout the rest of the article.

**L.320** The presence of both the high pressure over the Iberian and the low-pressure north of the British isles will be both almost mandatory requirements for a strong AR to landfall in the region of interest. However, I am not sure that the plots in Figure 11 are really catching the importance of the low-pressure system, which is essentially the one that is carrying the warm conveyor belt and the AR in its pre-frontal region. Particularly, I would be interested to know how many of those 10 systems were explosive cyclogenesis. There is a recent article (Eiras-Barca et al., 2018) analyzing the important correlation between explosive cyclogenesis and strong ARs over Europe, and it would be interesting to know how many of those 10 systems leading to the 10 highest flood peeks were explosive cyclones.

Additionally, I think that there is room here for a brief discussion about the role played by the NAO in all this. I suggest the authors include a brief discussion on the matter.

**Figure 1** Is not clear what "euro_dem" is.

**Figures 3,4,6,7,9,10,11** Please include the units in the colorbars or the arrows.

---

## Author Comment (AC1) · 29 Jun 2020

R: We thank the reviewer for the suggestions/comments/feedback that helped us improve our manuscript and for tacking time to read and review our paper.

The paper provides an interesting and unique contribution showing atmospheric rivers as drivers of high flows in the lower Rhine catchment. It thus enriches our understanding of hydrometeorological flooding drivers at the catchment level for this global region. By utilizing a long and comprehensive meteorological record, the authors show how indeed ARs have led to important damages in the region. It also provides interesting insights these events are preceded up to 7 days by intense moisture transport from the tropical North Atlantic basin typically precede ARs. The comments I have are minor which concern mainly methodological clarifications as well as suggestions to provide more insights of the repercussions of their findings. If a new version of

the manuscript successfully address these issues I would recommend the article for publication. Please find here below my specific observations:

Line 78: "i) analyze from a hydrological point of view" This sounds rather broad and ambitious; this line should be refined to specify what 'hydrological' actually means in the context of this publication

R: We will modify the text following the reviewer's suggestion.

Line 81: "iii) link the flood peaks with 80 the occurrence of AR". It does not look very scientifically sound to try to find a link between these two aspects. This suggest that the objective is already anticipating the results. This objective should be changed to something along the lines of 'explore how the occurrence of ARs explains flood peaks' or similar

R: Thank you for this comment. We will change the paragraph according to reviewer's suggestion.

95-102: This paragraph should be enriched with references. It sounds as if data has been already processed and the authors are already presenting their results

R: We will modify the text following the reviewer's suggestion. More references will be added.

Line 98: "winter or spring floods, which are triggered by warm air intrusions with corresponding snow melt in flatlands and low mountain ranges and summer floods, which are fed by large-scale 100 heavy rain or long-lasting repeated precipitation episodes (in connection with late snowmelt / glacier runoff in the Alps)" Please improve the fluidity of this sentence or try to split it two. At present it is not very clear

R: The paragraph will be modified and improved in the revised version of the manuscript.

Line 112: This paragraph would be enriched by a short 1-2 sentence conclusion about

the general hydrological trends of these tributaries.

R: We will add the required in formation in the revised version of the manuscript.

Line 114: How is this station representative for the catchment? What about the impact of upstream water infrastructure (dams, reservoirs, levees, and others which may mitigate floods)? Wouldn't readings at this specific part of the catchment give a misleading interpretation if the catchment is heavily intervened? Is hydraulic and infrastructure intervention indeed important here?. I think that this is very relevant when you compare events of 1925 vs the 1990s. I suppose that a number of hydraulic interventions have been made in the river in a period of ∼60 years. Probably these interventions and the fact that you are just looking at river gauge readings are leading us to underestimate the connection between ARs and flood peaks. Even if the role of hydraulic infrastructure is not deemed as relevant in this part of the catchment, it would be useful to clarify these aspects throughout the text.

R: In the revised version of the manuscript we will add a paragraph regarding the importance of Köln gauging station as well as the influence of river training on the flood magnification in the Lower Rhine. Although Rhine River has been intensively modified, most of the river training was don in the upper Rhine, especially on the Swiss side of the river. There are numerous studies showing that the modification on the catchment area have a relatively small influence on the flood peak in the Lower Rhine. All this information will be added in the data and methods part in the revised manuscript.

Line 144: reference? What is the general proportion of floods happening during winter vs summer? Are winter ones more or less frequent (generally speaking although these trends, naturally, are not constant)?

R: We will add a table with the occurrence rate of annual maxima over the period 1817 – 2019. The annual maxima, @Köln gauging station occurs mainly during the winter months. This information will be added also in the revise version of the manuscript.

Line 214 and in general throughout the text: The manuscript would be highly enriched if somehow these monetary losses are translated to current usd/eur values; not asking the authors to perform a complex econometric calculation but it would be useful to put these economic losses in perspective. For example, the 1925 event seemed to have caused losses of 100 Million DM vs 50 Million DM in 1993 vs 500M is 1995. How do they compare each other nowadays in current USD/EUR?. Similarly, the text would be enriched if the authors provide a table (or figure) comparing the impacts that these events have had on human lives, displaced people, monetary losses, infrastructure damages (even a qualitative description), and others. This would provide a useful information to understand the truly impacts of ARs in this key global catchment.

R: We agree with this comment and we will try to change the currency of monetary losses in Euro. We can make mostly just a rough approximation, because we do not have data regarding the inflation rate for 1925/26 for example.

Line 333, Conclusions:

o In general I think that either here or in previous sections there should be a short discussion describing the general trends of ARs-caused high flows over these 2-century time period. With the data you already have, it would be very useful to have a perspective on whether ARs-caused floods in the Rhine have been more recurrent? Or more intense? Both? None? While I understand that a full and comprehensive trend type of analyses might be out of the scope of this study, the manuscript will be highly enriched if even few sentences are added exploring this issue.

R: We fully agree with this comment and we will try to improve the conclusion part in the revised version of the manuscript tacking into account the aforementioned suggestion.

The conclusions should also highlight the socio-economic impacts that these events have caused.

R: This information will be added to the conclusion part in the revised version of the

manuscript.

Please also note the supplement to this comment:
https://www.hydrol-earth-syst-sci-discuss.net/hess-2020-149/hess-2020-149-AC1-supplement.pdf

---

## Author Response (AR1)

**Reviewer 1**

**R: We thank the reviewer for the suggestions/comments/feedback that helped us improve our manuscript and for tacking time to read and review our paper. Below you will find a point by point response to each concern/request raised throughout the review process.**

The paper provides an interesting and unique contribution showing atmospheric rivers as drivers of high flows in the lower Rhine catchment. It thus enriches our understanding of hydrometeorological flooding drivers at the catchment level for this global region. By utilizing a long and comprehensive meteorological record, the authors show how indeed ARs have led to important damages in the region. It also provides interesting insights these events are preceded up to 7 days by intense moisture transport from the tropical North Atlantic basin typically precede ARs. The comments I have are minor which concern mainly methodological clarifications as well as suggestions to provide more insights of the repercussions of their findings. If a new version of the manuscript successfully address these issues I would recommend the article for publication. Please find here below my specific observations:

• Line 78: "i) analyze from a hydrological point of view" This sounds rather broad and ambitious; this line should be refined to specify what 'hydrological' actually means in the context of this publication

**R:** The text has been modified accordingly.

• Line 81: "iii) link the flood peaks with 80 the occurrence of AR". It does not look very scientifically sound to try to find a link between these two aspects. This suggest that the objective is already anticipating the results. This objective should be changed to something along the lines of 'explore how the occurrence of ARs explains flood peaks' or similar

**R:** Thank you for this comment. We have modified the paragraph according to reviewer's suggestion.

• 95-102: This paragraph should be enriched with references. It sounds as if data has been already processed and the authors are already presenting their results

**R:** Following the reviewer's suggestion we have added more reference for this paragraph and extended with more information.

• Line 98: "winter or spring floods, which are triggered by warm air intrusions with corresponding snow melt in flatlands and low mountain ranges and summer floods, which are fed by large-scale 100 heavy rain or long-lasting repeated precipitation episodes (in connection with late snowmelt / glacier runoff in the Alps)" Please improve the fluidity of this sentence or try to split it two. At present it is not very clear

**R:** The paragraph has been modified and improved in the revised version of the manuscript.

• Line 112: This paragraph would be enriched by a short 1-2 sentence conclusion about the general hydrological trends of these tributaries.

**R:** We have added the required in formation in the revised version of the manuscript.

• Line 114: How is this station representative for the catchment? What about the impact of upstream water infrastructure (dams, reservoirs, levees, and others which may mitigate floods)? Wouldn't readings at this specific part of the catchment give a misleading interpretation if the catchment is heavily

intervened? Is hydraulic and infrastructure intervention indeed important here?. I think that this is very relevant when you compare events of 1925 vs the 1990s. I suppose that a number of hydraulic interventions have been made in the river in a period of ~60 years. Probably these interventions and the fact that you are just looking at river gauge readings are leading us to underestimate the connection between ARs and flood peaks. Even if the role of hydraulic infrastructure is not deemed as relevant in this part of the catchment, it would be useful to clarify these aspects throughout the text.

**R:** In the revised version of the manuscript we have added a paragraph regarding the importance of Köln gauging station as well as the influence of river training on the flood magnification in the Lower Rhine. Although Rhine River has been intensively modified, most of the river training was don in the upper Rhine, especially on the Swiss side of the river. There are numerous studies showing that the modification on the catchment area have a relatively small influence on the flood peak in the Lower Rhine.

• Line 144: reference? What is the general proportion of floods happening during winter vs summer? Are winter ones more or less frequent (generally speaking although these trends, naturally, are not constant)?

**R:** We have added a new figure (Figure S1) with the occurrence rate of annual maxima over the period 1817 – 2019. The annual maxima, @Köln gauging station occurs mainly during the winter months. This information has been added in the revised version of the manuscript. More than 80% of the annual maxima of daily streamflow at Köln gauging station occur during the extended winter months (November – March).

• Line 214 and in general throughout the text: The manuscript would be highly enriched if somehow these monetary losses are translated to current usd/eur values; not asking the authors to perform a complex econometric calculation but it would be useful to put these economic losses in perspective. For example, the 1925 event seemed to have caused losses of 100 Million DM vs 50 Million DM in 1993 vs 500M is 1995. How do they compare each other nowadays in current USD/EUR?. Similarly, the text would be enriched if the authors provide a table (or figure) comparing the impacts that these events have had on human lives, displaced people, monetary losses, infrastructure damages (even a qualitative description), and others. This would provide a useful information to understand the truly impacts of ARs in this key global catchment.

**R:** We agree with this comment and we have tried to change the currency of monetary losses in Euro. We were able to make mostly just a rough approximation, because we do not have data regarding the inflation rate for 1925/26 for example.

• Line 333, Conclusions:

o In general I think that either here or in previous sections there should be a short discussion describing the general trends of ARs-caused high flows over these 2-century time period. With the data you already have, it would be very useful to have a perspective on whether ARs-caused floods in the Rhine have been more recurrent? Or more intense? Both? None? While I understand that a full and comprehensive trend type of analyses might be out of the scope of this study, the manuscript will be highly enriched if even few sentences are added exploring this issue.

**R:** We fully agree with this comment and we have changes substantially the conclusion part. We have added also some discussion in the section to make it more readable.

o The conclusions should also highlight the socio-economic impacts that these events have caused.

**R:** As stated before the conclusions part has been modified substantially and we have tried to take into account all reviewer's suggestions/comments.

**Reviewer 2**

**We thank the reviewer for the suggestions/comments/feedback that helped us improve our manuscript and for tacking time to read and review our paper. Below you will find a point by point response to each concern/request raised throughout the review process.**

Review: Rivers in the sky, flooding on the ground

This paper investigates the relationship between atmospheric rivers and extreme flooding in the lower Rhine basin. So far, studies mostly investigated this relation between extreme rainfall (or flooding) and ARs for coastal regions. Therefore, the objectives to analyze the connection between ARS and floods occurring more inland is relevant and interesting, and fits the scope of HESS. The study describes the hydrometeorological situation of the three most extreme flood events over the lower Rhine basin in the last 180 years, and the connection with atmospheric rivers. In addition, composites of the large-scale circulation and IVT describing atmospheric rivers are analyzed for the 10 most extreme flood events. Although the analyses answer the research objectives, in my view there is much more potential and knowledge to gain from the dataset and method then is done so far in the manuscript. In fact, the section on the composites of the 10 largest flood events is very short, and has much more potential. I will give a few suggestions on additional questions/ experiments, and leave it up to the author/editor if these analyses are needed in this manuscript or can be potentially used as research questions for further studies.  In addition, I have some major comments and unclarified on the paper below, which should be addressed before publishing. Furthermore, the writing of the manuscript can be improved, being consistent in tense, English language, and use of units throughout the manuscript. Please find my minor comments at the end of this document.

Additional questions arising from the manuscript:

How are the trends in flooding and ARs over the dataset? The dataset spans quite a substantial time (1836-2015) and in the method section it is indicated that floods probably would happen more frequently (Line 101-110), so it would be nice to explore this further using this dataset. You can also assess if Atmospheric Rivers are going to be more frequent and intense over the lower Rhine catchment as you refer to that is the case over the North Atlantic (line 370)

**R:** In the revised version of the manuscript we have included some text and a new figure (Figure S14) showing the trends in the annual maxim at Köln gauging station. Regarding the ARs  we not aware of any observational study that has shown a robust and prominent trend in ARs in the historical record. Nevertheless, we have tried to compute the number of days when an AR occurs over the lower Rhine and add this information in the manuscript (see last paragraph of section 3.4).

ARs are projected to increase in the future but analysis of the historical trend, if observable, is challenging given the limited length of ground truth for ARs and complication from multi-decadal natural variability. Regarding the future projections, we can just rely on already published studies, because an in-depth analysis regarding this topic is beyond the scope of the current study.

As Atmospheric Rivers are associated with the Warm Conveyor Belt of extratropical cyclones, besides holding a lot of moisture, temperatures are often warmer than normal within these ARs. The effect of temperature on snow melt in the Alps or lower Rhine basin could positively influence discharge peaks. This aspect remains underexposed in this study. For example in the case of high temperature inducing snow melt could be related.

**R:** We fully agree with this comment. Most of the flooding for our selected events were related to a sharp increase in the temperature which led to snowmelt. We have added more information/ discussion regarding this issues in the revised version of the manuscript.

> I miss the explanation of the mechanism how Atmospheric Rivers result in extreme rainfall (for coastal areas; when an AR reaches a topographical barrier air parcels are lifted and adiabatically cooled, resulting in clouds and precipitation). For the lower Rhine this lifting mechanism is probably related to the Ardennes area? I am wondering if these mountains trigger enough uplift to result in precipitation or that the amounts of moisture during the three investigated cases is so high that only little uplift is needed.

**R:** The extreme flooding events at Köln and in the lower Rhine area are mainly related to extreme flooding of the Moselle river. This is one of the main reason for which we focused our manuscript also on the streamflow and precipitation data at Trier station (situated ion the main course of Moselle river). The catchment area of Moselle river includes the western side of the Vosges Mountains (elevation ~ 1,424 m) and the southern part of the Eifel range. We have added this information and more physical explanation regarding the extreme rainfall and ARs over our analyzed region, in the revised version of the manuscript.

> You have selected the 3/10 most extreme flood events in the lower river Rhine area and linked those to Atmospheric Rivers. In a dataset of 180 years, more flood events could be selected and their link with Atmospheric Rivers can be investigated. With that, the robustness of the link between ARs and extreme rainfall and flooding over the lower Rhine basin can be analyzed and put into perspective.

**R:** We agree with this comment, but our aim was to focus on the most extreme floods in the lower Rhine. For the current study we will keep just this 10 flood events, but in the future we want to make another study in which we want to include more floods events and also analyzed the floods and their triggers in the upper Rhine area.

**Major comments:**

The title is illustrative, although does not exactly reflect the novelty of this research with the focus on the link between large-scale circulation (ARs) and floods **inland**.

R: We agree with this comment. We have modified the title of our manuscript as follows: "Rivers in the sky, flooding on the ground: the role of atmospheric rivers for the inland flooding in central Europe"

Atmospheric studies such as Helen Dacre (2015) argue that atmospheric rivers are a result of constant local recycling of water along the cold front of an extratropical cyclone. This suggests that precipitation related to ARs originates more locally rather than from sub-tropical regions as is indicated in line 68, 74, 336 etc. Please discuss or revise.

**R:** The text has been modified accordingly. We appreciate the pointer to the Dacre (2015) study, but would like to note that the role of local versus remote moisture sources in ARs likely varies from region to region and from cases to cases. It's not universally true that "precipitation related to ARs originates

more locally rather than from sub-tropical regions". We have tried to modify the text in such a way not to put extensive focus on the tropical source for the moisture transport.

In the introduction ARs and the relation with different teleconnections is mentioned, but I don't see that coming back in the rest of the paper. Have you checked NAO indices for the events you studied? This could be interesting, as you often refer to a low south of Greenland (Iceland) and a high over the coast of North Africa (Azores), which gives indication for a positive NAO resulting in strong flows to northwestern Europe. It would be interesting to invest the index of NAO for the selected flood events.

**R:** A discussion regarding NAO has been added in the revised version of the manuscript (see Section 3.5). For the period 1950 – 2019, when we have access to daily NAO values, NAO was in a positive case for the flood events of 1970, 1988 and 1993. For the flood event of 1995, NAO was altering between positive and negative values the week preceding the flood event. One of the most extreme case was for the December 1993 flood, when NAO was in positive phase from 16. October until 22 December 1993.

Although your showing IVT in the figures, I miss the embedding in the text. The case studies could have more focus on this IVT values and how anomalous they are, as to my knowledge values in IVT of 800 kg/m/s are quite exceptional. It would be nice to put this a bit more in perspective.

**R:** We fully agree with this comment and we have modified the text accordingly in the revised version of the manuscript, in such a way to integrate the IVT values better throughout the text.

The lag between AR/extreme precipitation is interesting and could deserve some more attention throughout the manuscript. Of course this mainly depends on local timing and location, indicating the importance of local processes (to be added on line 40) although it depends on the size of your catchment. Both in the conclusion as in the abstract I miss quantification of the results. Can you give some numbers to align your statements? For example indicate how anomalous the selected events were in terms of discharge and IVT related to the AR. I think an important conclusion from this research is that in these extreme events, large-scale circulation are rather similar but local conditions leading to the flood not. This is an interesting conclusion which could be highlighted more, and could be strengthened if it was further quantified in the conclusion and abstract as well.

**R:** Following the reviewer's comment/suggestion we will modify the text in such a way to add more information regarding the magnitude of the selected events and how anomalous they are in a long term perspective. We will change the text through the entire manuscript to add the required information where is needed.

**Minor comments:**
Line 8: The role of the large scale atmospheric circulation
Line 13: sentence: The influence of ... In my view atmospheric rivers are part of the large-scale circulation, the .. is done via the prevailing large-scale atmospheric circulation.. sounds very odd to me
Line 34: that coping with floods is not **trivial**
Line 40: I would argue that for flood forecasting and the time of occurrence, location and magnitude scales from mesoscale to local scale are important. Especially knowledge on local orography is needed to get the location of the flooding right
Line 42: Same sentence as last sentence of previous paragraph.. combine?
Line 49: This sentence is not clear. What do you mean with regional and local climates?
Line 115: Rhein --> Rhine

Line 118: what is the time resolution of the reanalysis data?

Line 119: I am confused that a re-analysis dataset has ensemble members, could you explain that?

Line 121: has several improvements > vague statement, either name improvements or leave out

Line 133: divided by gravity (g).

Lines around 133: What is the vertical resolution of your wind and specific humidity data?

Line 152: Where do you show the EOBS gridded precipitation dataset? Not clear how and if this comes back in your results. Although it would be good to analyze a gridded precipitation field over the lower Rhine basin instead of a point observation at Trier.

Line 161: .. in the large parts of the Rhine catchment..

Line 164: od -> of

Line 167: Can you quantify the total amount of precipitation over the lower Rhine basin area instead of showing the gridpoint values in the appendix. That would give more quantification to the results.

Line 169: This sounds like a positive North Atlantic Oscillation. Would be good to check the index for the selected events.

Lines around 170: Miss numbers of IVT in 1925 case, that would give indication of the 'severity' of AR. Can you give some IVT values here to give some indication as you do for the hydrometeorological situation of 1993 (line 229). In general it would be good to embed the values of IVT a bit more within the text as I think they are quite exceptional, can you compare with climatology? Or average value of ARs at this latitude (as the value of IVT is latitude dependent)

Line 180: specific humidity or moisture

Line 183 etc: Why do you talk about wind and moisture separately here? You can refer to IVT and Figure 3 and in my opinion there is no need in showing figure 4 as it gives the same information as Figure 3.

Line 223: too should be to

Line 228: Again, could you give precipitation values averaged over the basin? And compare that to climatology?

Line 238: Where is statement based on? Add reference

Line 268: become > became. Keep tenses consistent throughout the result section.

Line 268: Where is Berus meteorological station located? Indicate in map

Line 298: driver > driven

Line 319: plum should be plume?

Line 321: by a south-westerly wind (Figure 11) > you are showing IVT vectors and no wind vectors in Figure 11 so wrong reference

Line 322: By visual inspection.. etc. Are you referring to the individual ARs per event or the composites here? If you refer to the composites I would not expect to see individual ARs as the composite gives an average and the ARS are therefore smoothed and you can expect IVT with widths bigger than 1000 km wide.

Line 336: Sentence 2 in the conclusion is not a conclusion of your work, but new information: I would move it to the introduction

Line 346: what is meant by westerly, southwesterly and north-westerly large-scale circulation types? Do you mean prevailing winds? Please clarify.

Line 358-363: This sentence is an explanation why more storms (ARs) occur in winter and should be moved to the methodology section to explain why this study focuses on wintertime.

Line 363: .. is done .. this sentence is not very easy to read

Line 379: I guess this research is about the UK? Maybe good to add that in the sentence to be more specific

**R:** All the minor comments/suggestions have been taken into account and addressed individually in the revised version of the manuscript.

**Table & Figures**

I see in Table 1 that the magnitude of the flood of 1995 is as high as the one in 1920, so why was the case of 1995 described and not the one from 1920? Or is it because these stream flows are from Koln while you based your analysis on the ones from Trier? This should be stated more clearly in the text, as it is not clear from the methods.

**R:** The choice of 1925/26, 1993 and 1995 is mainly due to data availability (precipitation at Trier gauging station, daily precipitation over Germany - REGNIE database). We have added this information in the revised version of the manuscript.

Also streamflow in Trier station is mentioned in Figure 4 and similar figures, while I understand from the Composite events section that you base your analysis on flood peaks measured at Koln (should be Cologne) gauging station. This is confusing.

**R:** We will modify the text in such a way to make more clear why we used both flood peaks at Trier and Köln. In general, the floods peaks at Köln are preceded by a few days by flood peaks at Trier gauging station. Most of the flood event at Köln and in the lower Rhine are triggered by flood peaks on the Moselle River (where Trier gauging station is situated). That's why we show both time series.

Missing units in Figure 3, 4 and all similar figures

**R:** The units have been added for all the figures in the manuscript in a proper manner.

In my opinion the figures with daily specific humidity and wind at 900 hPa do not add enough additional information to be shown in the main manuscript.

**R:** These figures have been removed from the revised version of the manuscript.

Figure 2: alone should be along

**R:** Modified as suggested.

Figure 2b and all subsequent figures. Is it needed to show daily streamflow from all these locations and for the whole year? If so, those locations should also be located in Figure 1 or mentioned in the methods section. In my opinion the a figures with discharge at Trier and Cologne give enough information and I would rather show the spatial distribution of precipitation as an addition.

**R:** We agree with the reviewer, but the main message by using all gauging station on Rhine catchment area is to show that the flood peaks at Köln gauging station are mainly influenced by the flood waves of the Moselle river and to a much lesser extend by the streamflow in the middle and upper Rhine basin.

Figure 12: Not sure what the colors present here? Are these the colours for the 10 different extreme events? I cannot imagine that the orange AR just south of Greenland at Lag0 leaded to a flooding in the Alps, can you comment? This figure needs more explanation in the caption and also a color scale.

**R:** In Figure 12 there is a representation of all ARs recorded, over the whole North Atlantic basin, prior to the floods. Some of the ARs do not reach the European continent and are not related to our analysis, but is was very hard to exclude them from the figure. It is a rather challenging technical problem due to the data format. Nevertheless, we have tried to exclude all the ARs which are not related to our floods from the figure in the revised version of the manuscript.

Some figures in the additional material miss units and the data sources should be clarified in more detail, are these gridded observation data, re-analysis?

**R:** All figures have been modified/improved following all the comments/suggestions from all 3 reviewers.

**Reviewer 3**

**R: We thank the reviewer for the suggestions/comments/feedback that helped us improve our manuscript and for tacking time to read and review our paper. Below you will find a point by point response to each concern/request raised throughout the review process.**

**Rivers in the Sky, flooding on the ground**
**Reviewer 3 Report Round 1**

This article analyzes the role played by atmospheric rivers in some of the most important flood events in the lower part of the Rhine River basin. Overall, the paper is well written, and the inclusion of the perspective of the hydrological extremes –floods– rather than the simple extreme precipitation is always an added value. The authors find most of the more important flood events over the region were preceded by an AR event, and this is an interesting result that could be valuable for the region. The quality of the figures is acceptable, but it could be improved. I suggest the authors improve some of them if that is not very problematic.
I believe that the title is a bit pretentious. It is a very catching title that would be probably the best choice for a review paper or a paper intended to get conclusions on a global scale. This manuscript is focused on a very particular region of Inland Europe, and I think that this should be reflected in the title somehow. I would perfectly understand if the authors would like to keep the "Rivers in the sky, flooding on the ground" –I would have done the same–, but I suggest that this title should be extended with a citation to the region of interest somehow.

R: We agree with this comment. We have modified the title of our manuscript as follows: "Rivers in the sky, flooding on the ground: the role of atmospheric rivers for the inland flooding in central Europe"

I have already read the comments made by the other reviewers, and I mostly agree with them. Reviewer 2 suggests to extend the 10-events composites. I will not put that condition as necessary to give my full recommendation to publish, but I think that it could be a good improvement for the paper if the authors are willing to do it.

**R:** Although we fully agree with this comment/suggestion, our aim was to focus on the most extreme floods in the lower Rhine. For the current study we will keep just this 10 flood events, but in the future we want to make another study in which we want to include more floods events and also analyzed the floods and their triggers in the upper Rhine area. If we extend the current study to more flood events, it means changing almost substantially the structure and the outcome. We want to regard this study as a starting point for more in-depth studies regarding extreme inland flooding and their large scale driver, with a special emphasis on ARs.

Also, this colleague suggests the authors include a discussion about Helen Dacre's (and others) perspective of the importance of local convergence of moisture in ARs development. He/She is right, but I would like the authors to take into account –when they discuss this point– that there is also a huge bunch of articles of all kinds pointing out to the essential role played by the large scale advection of tropical and subtropical moisture. I do not think that the authors should take sides with any of those perspectives –actually, I believe that both mechanisms are necessary, and the relative importance between them changes among the events–, but both may be included in the discussion.

**R:** We agree with both reviewers comments. Thus, we have tried to change the text in the revised manuscript to be able to properly discuss and integrate the aforementioned comments/concerns. Some parts of the revised version of the manuscript have been substantially modified and new information has been added.

I will not suggest major changes, however, some of my comments (particularly those regarding the very likely explosive nature of some of the involved cyclones and also those regarding the role played by NAO) will take some time from the authors to be replied. I would like to read and discuss the answers in an eventual second round of the review process.

**R:** We have tried our best to improve the revised manuscript tacking into account the reviewer's suggestions/comment.

**Minor Comments**

**L.43** I suggest the authors consistently arrange the citations by chronological order. It is not only fairer for our colleagues, but also the result is more elegant. For example, in this case, I would start from Lavers and finish by DeFlorio or Guan and Waliser.

**R:** The references have been modified accordingly.

**L.54** Please, leave a blanck space between "50" and "km".

**R:** Modified as suggested.

**L.57** The beneficial aspects of ARs are not restricted to arid/semiarid areas at all. Most ARs are beneficial even in mid-latitudes.

This idea is well discussed in Ralph (2019), and I think that should be included somehow in the text. Ralph, F. M., Rutz, J. J., Cordeira, J. M., Dettinger, M., Anderson, M., Reynolds, D., ... & Smallcomb, C. (2019). A scale to characterize the strength and impacts of atmospheric rivers. Bulletin of the American Meteorological Society, 100(2), 269-289.

**R:** Modified as suggested.

**L.65** There are some other important analyses relating ARs and extreme precipitation and floods in Europe that the authors did not take into account. (e.g. Eiras-Barca et al.; 2016, 2017).

**R:** We have integrated the aforementioned references in the revised version of the manuscript.

**L.87** I think that section 2 must be included in the methods section. However, this is just my opinion and I let this decision to the authors.

**R:** We agree with this suggestion and we moved section 2 in the data and methods section.

**L.116** If the authors had SLP, why did they start the vertical integration at the level of 1000 hPa instead of SLP, which would the most correct option?

**R:** We start the vertical integration at 1000 hPa due to the limitation for the wind and specific humidity data. The SLP data is used to look at the large scale circulation. For the vertical integration we are actually using the surface pressure.

**L.130-134** I don't see the need to describe with words what the equations are already saying.

**R:** Modified as suggested.

**L.135** The algorithm (and database) developed by Guan at UCLA is one of the most commonly used in our field, and I am not going to call it into question. However, did the author consider the possibility that the detection thresholds could have substantially changed in these almost 200 years that they are considering in the analysis?

**R:** We appreciate this interesting question. Fundamentally, this relates to the question whether the definition of ARs should change as the climate changes, and we think it could be argued either way. In the current study, we chose to use a fixed AR definition – similar approaches have been used for studying AR changes between the current and future climates (e.g., Espinoza et al. 2018; Massoud et al. 2019) and have proved useful. This is also consistent with the study of other types of extremes. For example, a hurricane a century ago is still a hurricane today.

**L.136** Please, replace the asterisks by ·

**R:** Modified as suggested.

**L.152** How well is performing EOBS over Germany? Some analyses pointed out the fact that EOBS may not be the best option over continental Europe...

**R:** The new EOBS version (v21.e) is rather robust over Germany, mainly because in the E-OBS database the highest number of station are the German ones. The E-OBS data was included in the supplementary figures mostly to show that the precipitation was not located over Germany, but it was stretching From France towards Germany in a rather narrow band. In the revised version of the manuscript we have included also the REGNIE dataset, which is a German product of daily precipitation with a 1km x 1km spatial resolution and which is restricted only to country level.

**L.164** Please, leave a blank space between 27 and mm. Take this into account throughout the rest of the article.

**R:** Modified as suggested.

**L.320** The presence of both the high pressure over the Iberian and the low-pressure north of the British isles will be both almost mandatory requirements for a strong AR to landfall in the region of interest. However, I am not sure that the plots in Figure 11 are really catching the importance of the low-pressure system, which is essentially the one that is carrying the warm conveyor belt and the AR in its pre-frontal region. Particularly, I would be interested to know how many of those 10 systems were explosive cyclogenesis. There is a recent article (Eiras-Barca et al., 2018) analyzing the important correlation between explosive cyclogenesis and strong ARs over Europe, and it would be interesting to know how many of those 10 systems leading to the 10 highest flood peeks were explosive cyclones. Additionally, I

think that there is room here for a brief discussion about the role played by the NAO in all this. I suggest the authors include a brief discussion on the matter.

**R:** Following the reviewer's suggestion/comment we have actually checked if some of the extreme floods analyzed in our study (e.g. 1988, 1993 and 1995) are also associated with explosive cyclones. We have used the database kindly provided by Jorge Eiras-Barca (https://esd.copernicus.org/articles/9/91/2018/esd-9-91-2018.html). For the aforementioned 3 extreme flood events no explosive cyclones have been recorded during the flood peaks or prior to them.
For the whole analyzed period (1836 – 2019) it has been shown that the NCEP-20C reanalysis dataset as well as ERA-20C data are not optimal datasets for extratropical cyclones and windstorms analysis (Befort et al., 2016) (https://rmets.onlinelibrary.wiley.com/doi/full/10.1002/asl.694). In their study Befort et al. (2016) have show that the use of the long-term reanalysis dataset (NCEP- 20C and ERA-20C) is hampering a reliable analysis of real long-term trends of cyclone and windstorm activity. In a similar study Wang et al. (2013) (https://link.springer.com/article/10.1007/s00382-012-1450-9) have shown that the use of NCEP – 20C ensemble-mean is found to be unsuitable for accurately determining cyclone statistics. Thus, we cannot make a proper analysis regarding explosive cyclones and extreme flood events over Lower Rhine over the whole length of our dataset. Nevertheless, we have added a new section regarding the role played by the NAO in the occurrence of the flood events in the Lower Rhine (see Section 3.5)

**Figure 1** Is not clear what "euro_dem" is.

**R:** The text has been modified following the reviewer's suggestion.

**Figures 3,4,6,7,9,10,11** Please include the units in the color bars or the arrows.

**R:** The units have been added for all the figures in the revised manuscript in a proper manner.

[revised manuscript text omitted]

---

## Referee Report (RR1)

I would like to thank the author for addressing the issues raised in the previous review. I think the paper is improved and the additional analyses on the NAO in relation to AR, and the change in AR over the past century have given more 'body' to the research. That said, I think the text can still be improved and at some places shortened so the article becomes more to the point, and easier to read. Please find my minor comments on these issues below.

**Minor comments**

Line 25-40: The introduction starts very broad, and in my opinion you can start narrowing down the research a bit more in the introduction already so the purpose of the study becomes clearer at an earllier stage in the text. For example, is it needed to talk about floods in the UK and in the eastern part of Europe when you focus on the Rhine? Similar, is it needed to name the Pacific North American Oscillation when you focus on Europe? I think the start of the Introduction can be a bit more to the point.

Line 50: In line 50 you mention for the first time AR, where the explanation of what is an AR follows a bit later. This is an illogical order.

Line 60: what do you mean with 'can be visible'? From satellites?

Line 69: Sentence on New Zealand can be left out as it does not directly relate to the study here

Section 2.1 Catchment area: There is a lot of information provided in this section. I am not sure if all information is needed for this study. Please consider this.

Line 109-112: Line needs reference

Line 124: The overall increase in  winter precipitation

Section 2.2 Data: I suggest a re-ordering of this section as you now discuss discharge data, than re-analysis data/meteorological data, and then discharge data again. I suggest to move line 169-185 after line 145, to make the order more logic. Then you start with discharge data, then precipitation data, and then the larger-scale meteorological data.

Line 146: At which height did you extract zonal and meridional wind? Please add

Line 146-148: Did you extract both surface pressure and daily sea level pressure?

Lines 155: IVT is introduced and then in section 3.1 there is only referred to IWV, this variable should then also be introduced in this data section

Line 161: gravity --> gravitational constant

Line 210: Sentence is difficult to read

Line 211: third highest value, since when?

Line 218: IWV is not introduced, neither in the data section, please add

Line 304: Rockenau level and Kaub level? do you mean station instead of level?

Line 463: In this study we have shown that extreme floods.. --> indicate your criteria used for extreme floods.

Line 473-479 and Line 481-485: Both are very long sentences, which are hard to read. Please split the sentence in smaller sentences.

Line 498: ..anomaly than its positive counterpart --> not clear what you mean here with positive counterpart? Do you mean high pressure?

Line 498: .. and it guides the IVT is a narrow band through.. sentence should be improved

Table 2: Improve caption. Sentences are illogical now.

Figure 3 and similar: these figures contain a lot of information, and therefore also become a bit hard to read. A suggestion to improve the lay-out of this figure is to make the continents light gray and remove the countries and continental boundaries in black, as these overlap with the moisture arrows.

Figure S14a. This graph is interesting and relevant and in my opinion can be shifted to the main article to verify the findings on the trend in ARs.

---

## Author Response (AR2)

**We thank the reviewer for the suggestions/comments/feedback that helped us improve our manuscript and for tacking time to read and review our paper. We are happy that we were able to address in a proper manner the issues raised by the reviewer in the previous version of the manuscript . Please find below a detailed answer to the minor comments for the second round of review.**

I would like to thank the author for addressing the issues raised in the previous review. I think the paper is improved and the additional analyses on the NAO in relation to AR, and the change in AR over the past century have given more 'body' to the research. That said, I think the text can still be improved and at some places shortened so the article becomes more to the point, and easier to read. Please find my minor comments on these issues below.

**Minor comments**

Line 25-40: The introduction starts very broad, and in my opinion you can start narrowing down the research a bit more in the introduction already so the purpose of the study becomes clearer at an earlier stage in the text. For example, is it needed to talk about floods in the UK and in the eastern part of Europe when you focus on the Rhine? Similar, is it needed to name the Pacific North American Oscillation when you focus on Europe? I think the start of the Introduction can be a bit more to the point.

Resp.: We partially agree with this comment, but we have decided to keep the introduction as it is. The reason for which we start the introduction like this is to give an overview of the extreme floods that affected the European continent over the last decades and to emphasize that no studies currently exists which analyze in depth the extreme flooding in the lower part of Rhine River. I think we need a good argument why it is important to analyze floods and this is given in the first paragraph of the introduction.

Line 50: In line 50 you mention for the first time AR, where the explanation of what is an AR follows a bit later. This is an illogical order.

Resp.: We agree with this comment and we have modified the text in such a way to be more consistent (see lines 50 – 60)

Line 60: what do you mean with 'can be visible'? From satellites?

Resp.: We have modified the text to make it more understandable. We meant satellites.

Line 69: Sentence on New Zealand can be left out as it does not directly relate to the study here

Resp.: This sentence has been removed from the manuscript.

Section 2.1 Catchment area: There is a lot of information provided in this section. I am not sure if all information is needed for this study. Please consider this.

Resp.:We agree with this comment, but the reason for which we have such a detailed description of Section 2.1 was to fulfill the requirements of another reviewer. In the previous round of revision we were asked to be more detailed about the catchment area and to add more info regarding the river training along Rhine and why we use the Köln gauging station for this analysis.

Line 109-112: Line needs reference

Resp.: We have added a reference.

Line 124: The overall increase in the winter precipitation

Resp.: Modified as suggested.

Section 2.2 Data: I suggest a re-ordering of this section as you now discuss discharge data, than re-analysis data/meteorological data, and then discharge data again. I suggest to move line 169-185 after line 145, to make the order more logic. Then you start with discharge data, then precipitation data, and then the larger-scale meteorological data.

Resp.: Modified as suggested.

Line 146: At which height did you extract zonal and meridional wind? Please add

Resp.: We have added this information in the revised manuscript.

Line 146-148: Did you extract both surface pressure and daily sea level pressure?

Resp.: Yes we have used both. The daily sea level pressure is use to analyze the large scale atmospheric circulation, while the surface pressure is used to compute the IVT.

Lines 155: IVT is introduced and then in section 3.1 there is only referred to IWV, this variable should then also be introduced in this data section

Resp.: We have added the definition of IWV in the data section.

Line 161: gravity --> gravitational constant

Resp.: Modified as suggested.

Line 210: Sentence is difficult to read

Resp.: We have tried to modify the scented to make it more readable.

Line 211: third highest value, since when?

Resp.: We add this information in the revised version of the manuscript.

Line 218: IWV is not introduced, neither in the data section, please add

Resp.: We have added the definition of IWV in the data section.

Line 304: Rockenau level and Kaub level? do you mean station instead of level?

Resp.: We have modified the text according to the suggestion. It was station instead of level.

Line 463: In this study we have shown that extreme floods.. --> indicate your criteria used for extreme floods.

Resp.: We add this information in the revised version of the manuscript.

Line 473-479 and Line 481-485: Both are very long sentences, which are hard to read. Please split the sentence in smaller sentences.

Resp.: We have tried to modify the scented to make it more readable.

Line 498: ..anomaly than its positive counterpart --> not clear what you mean here with positive counterpart? Do you mean high pressure?

Resp.: We have changed the text to be more clear.

Line 498: .. and it guides the IVT is a narrow band through.. sentence should be improved

Resp.: We have changed the text to be more clear.

Table 2: Improve caption. Sentences are illogical now.

Resp.: The caption has been modified.

Figure 3 and similar: these figures contain a lot of information, and therefore also become a bit hard to read. A suggestion to improve the lay-out of this figure is to make the continents light gray and remove the countries and continental boundaries in black, as these overlap with the moisture arrows.

Resp.: We agree with the reviewer that some figures are having much information. Following the reviewer's suggestion we have made the countries light gray. We cannot remove completely the countries because it will be rather difficult to find the location of the study site, but we think that by making the border in light grey we removed a lot of useless information from the figures (se Figure3, 6, 9 and 11)

Figure S14a. This graph is interesting and relevant and in my opinion can be shifted to the main article to verify the findings on the trend in ARs.

Resp.: We have moved Figures S14 in the main article (now Figure 13).

[revised manuscript text omitted]